# Can we reconstruct the formation of large open ocean polynyas in the Southern Ocean using ice core records?

Hugues Goosse[1], Quentin Dalaiden[1], Marie G.P. Cavitte[1], Liping Zhang[2,3]

[1]Earth and Life Institute, Université catholique de Louvain, Louvain-la-Neuve, Belgium
[2]NOAA/Geophysical Fluid Dynamics Laboratory, Princeton, New Jersey, USA
[3]University Corporation for Atmospheric Research, Boulder, Colorado

*Correspondence to*: Hugues Goosse (hugues.goosse@uclouvain.be)

**Abstract.** Large open-ocean polynyas, defined as ice-free areas within the sea ice pack, have been observed only rarely over the past decades in the Southern Ocean. In addition to smaller recent events, an impressive sequence occurred in the Weddell
Sea in 1974, 1975 and 1976 with openings of more than 300,000 km$^2$ that lasted the full winter. Those big events have a huge impact on the sea ice cover, deep-water formation and more generally on the Southern Ocean and the Antarctic climate. However, we have no estimate of the frequency of the occurrence of such large open-ocean polynyas before the 1970s. Our goal here is to test if polynya activity could be reconstructed using continental records, and specifically, observations derived from ice cores. The fingerprint of big open-ocean polynyas is first described in reconstructions based on data from weather
stations, in ice cores for the 1970s and in climate models. It shows a signal characterized by a surface air warming and increased precipitation in coastal regions adjacent to the eastern part of the Weddell Sea where several high-resolution ice cores have been collected. The signal of isotopic composition of precipitation is more ambiguous and we thus base our reconstructions on surface mass balance records only. A first reconstruction is obtained by performing a simple average of standardized records. Given the similarity between the observed signal and the one simulated in models, we also use data assimilation to
reconstruct past polynya activity. The impact of open ocean polynyas on the continent is not large enough compared to the changes due, for instance, to atmospheric variability to detect without ambiguity the polynya signal and additional observations would be required to discriminate clearly the years with and without open ocean polynya. It is thus reasonable to consider that, in these preliminary reconstructions, some high snow accumulation events may be wrongly interpreted as the consequence of polynya formation while some years with polynya formation may be missed. Nevertheless, our reconstructions suggest that
big open ocean polynyas, such as the ones that were observed in the 1970s, are rare events, occurring at most a few times per century. Century-scale changes in polynya activity are also likely but our reconstructions are unable to assess precisely this aspect at this stage.

## 1 Introduction

Polynyas are ice-free oceanic areas within the sea-ice pack. They are regularly observed close to the coasts of Antarctica where
very strong winds coming from the continent push the sea ice away from the shore as soon as it is formed (Comiso and Gordon,

1987; Morales Maqueda et al. 2004). The open ocean polynyas, which are polynyas that occur far from the coast, are rarer and thus much less known. Several short-lived open ocean polynyas have been observed in the Southern Ocean over the last decades, with relatively large ones in the Weddell Sea in 2016 and 2017 reaching 50,000 km$^2$ (Comiso and Gordon, 1996; Swart et al., 2018; Jena et al., 2019; Campbell et al., 2019). In addition to those relatively short-lived events, the great Weddell

Sea polynya (Fig. 1) of 1974, 1975 and 1976 was truly exceptional in historical records, first by its size of 300,000 km$^2$, i.e. about 10 times the size of Belgium, but also because it remained open all winter long (Carsey, 1980; Zwally et al., 1983).

In contrast to the coastal polynyas, wind alone is not sufficient to maintain open ocean polynyas. A major oceanic heat source is required to sustain the large heat loss at the atmosphere-ocean interface and prevent sea ice formation (Morales Maqueda et al., 2004). In the Weddell Sea, this is achieved by open ocean convection that continuously brings warmer water from the

deeper oceanic layers to the surface (Gordon, 1978; Martinson et al., 1981). For the Weddell polynya of the 1970s, oceanic observations indicate mixing to a depth of 3000 m compared to a depth of about 100 m in normal years (Gordon, 1982). During the formation of the polynya in 2017, the observed mixing reached a depth of more than 1700 m (Campbell et al., 2019).

Although the role of deep oceanic mixing appears crucial, the specific mechanisms leading to the formation of open ocean polynya in the Southern Ocean is still under debate. Compared to other regions of the world, the stability of the water column

is low in the Southern Ocean (Gordon and Huber, 1984; Martinson, 1990). The cold and relatively fresh surface water is separated from warmer water at depth by a relatively weak pycnocline. This warm deep water is supplied by an input from the Antarctic circumpolar current (the circumpolar deep water), which itself originates in the deep water formed in the North Atlantic. Nevertheless, the low stratification is maintained by strong sea ice-ocean feedbacks (Martinson, 1990; Goosse et al., 2018; Wilson et al., 2019) and open convection reaching large depths is very rare.

Deep convection requires some preconditioning of the ocean, reducing the overall stability of the water column (Morales Maqueda et al., 2004; Dufour et al. 2017; Kurtakoti et al. 2018; Campbell et al., 2019). The opening of a polynya is then triggered by the winds and specifically by the passage of storms that export sea ice out of the region, enhance turbulent mixing in the ocean and may bring additional heat (Morales Maqueda et al., 2004; Cheon et al., 2015; Jena et al., 2019; Francis et al., 2019; Campbell et al., 2019). After the formation of the polynya by a particular event, the convection is self-sustained. The

warmer, saltier waters at depth are strongly cooled when they reach the surface by direct exchanges with the atmosphere, become denser and sink again to great depths. This convection also provides a preconditioning for subsequent years as it maintains a low stability of the water column, explaining why open ocean convection and polynya formation can be sustained over several years.

The great Weddell Sea polynya of the 1970s originated in the region close to a seamount called Maud Rise (at about 3°E,

64°S). This region is considered as particularly prone to polynya formation and short-lived polynyas are often observed there (Comiso and Gordon, 1996; Morales Maqueda et al., 2004). The main reason comes from interactions between the topography and the large scale circulation that lead to a shallowing of the mixed layer, upwelling of warmer deep water and generation of

mesoscale oceanic eddies inducing an overall reduced stability of the water column (Carsey, 1980; Comiso and Gordon, 1987; Holland, 2001; Kurtakoti et al., 2018).

The direct contact between the ocean at a temperature close to its freezing point and the very cold air above in open ocean polynyas in the winter has significant impacts on the atmosphere. The large turbulent fluxes induce an increase of the air temperature over the polynya area that can lead to a 20 degrees warming in winter compared to non-polynya years (Moore et al., 2002). The enhanced evaporation over the polynya induces a higher moisture content of the air, more clouds and more precipitation locally (Carsey, 1980; Moore et al., 2002; Weijer et al., 2017). The surface conditions may also cause a decrease

of sea level pressure over the polynya and thus influence the atmospheric circulation (Timmermann et al., 1999; Moore et al., 2002; Latif et al., 2013; Weijer et al., 2017; Kaufman et al., 2020). The local anomaly created over the polynya is transported downwind (e.g., Weijer et al., 2017), influencing the oceanic regions outside the polynya area, as well as the Antarctic continent. However, the signal there is less strong than over the polynya region and it is generally difficult to identify the effect of the polynya within the natural variability of the climate system (Carsey, 1980; Moore et al., 2002; Weijer et al. 2017).

Climate models have relatively large biases in their representation of vertical exchanges and deep water formation in the Southern Ocean (Heuzé et al., 2013; Sallée et al., 2013). A few models display intermittent open ocean convection that leads to the formation of polynyas covering a wide range of sizes and duration (Stössel and Kim, 2001; Martin et al., 2013; Zanowski et al., 2015; Weijer et al., 2017; Zhang et al., 2019; Kaufman et al., 2020). By contrast, widespread open ocean convection occurs nearly every year in some models (Manabe et al., 1991; Goosse and Fichefet, 2001; Heuzé et al., 2013; Stössel et al.,

2015) while other models have limited or no open ocean convection.

Unfortunately, the short instrumental records do not provide precise estimations of the frequency and the overall role of polynya formation and deep convection in the climate system nor can they be used to determine which climate models represent polynya occurrence adequately. It has been speculated that open ocean convection was more widespread in the past, with a reduction over the last decades that may have been caused by the large scale freshening observed in the Southern Ocean (de

Lavergne et al. 2014). Human-induced climate change will likely reduce further the probability of ocean convection in the future (de Lavergne et al., 2014; Heuzé et al., 2015, Kurtakoti et al., 2018). However, it is difficult to assess the magnitude of any recent change in polynya occurrence and their impact.

An option is then to study a time period that is longer than the one covered by instrumental observations and rely on the signal stored in natural archives. Unfortunately, to our knowledge, no high-resolution ocean sediment core that might provide a direct

record of polynya activity is available and, up to now, no reconstruction of polynya occurrence has been developed for the past centuries. This implies that the frequency of open ocean polynya formation is basically unknown. However, polynyas have an influence on the continent too and it might be possible to reconstruct their occurrence from a network of continental records. In that framework, ice core records are likely the best candidate as they provide high resolution, well-dated records of climate changes over the Antarctic continent.

Our goal here is to test if it is possible to reconstruct polynya activity using available ice core records, in particular the water isotopic composition ($\delta^{18}$O) and surface mass balance for which recent compilations have been developed (Stenni et al., 2017a; Thomas et al., 2017). The first step is to estimate, using modern observations and model results, where the signal is likely to be the clearest over the continent. This is done in Sect. 3. To leave an imprint in a natural archive, which generally has an annual resolution at best, the polynya must be large enough and stay open for a sufficiently long time. We thus focus on the major events such as the great Weddell polynya observed in 1974, 1975 and 1976. In Sect. 4, we then determine how the ice core data can constrain the evolution of polynyas over the past centuries using a very simple statistical technique and data assimilation. The final section (Sect.5) presents the conclusions of our analyses and discusses some perspectives for future developments.

## 2 Data and methods

### 2.1 Observations

To characterize the continental temperature changes occurring during the opening of the great Weddell Sea polynya in 1974, 1975 and 1976, we will first use direct observations from weather stations as well as a spatial reconstruction of temperatures based on these observations covering the period 1958-2012 (Turner et al., 2004; Nicolas and Bromwich, 2014a).

Measuring precipitation directly in Antarctica is much more difficult than temperature and many weather station records do not include this variable routinely (Turner et al., 2004). We will thus rely on a recent synthesis of surface mass balance (SMB) from 79 ice cores (Thomas et al., 2017). The surface mass balance is defined as the net surface accumulation resulting from precipitation minus removal from snow drift and sublimation, but it is mainly influenced by snow falls over Antarctica (e.g. Lenaerts et al., 2019). The ice core data provide direct but point estimates at the core locations. Additional information on the spatial structure of the changes during polynya formation can be obtained from a reconstruction of the surface mass balance (Medley and Thomas, 2019) that combines ice core data and atmospheric reanalysis fields in order to cover the whole grounded Antarctic ice sheet over the past 200 years. This combination has the advantage of using the spatial covariance represented in the reanalysis without the potential troubles associated with the lower quality of the reanalyses before 1979 and the inhomogeneities due to the inclusion of additional satellite observations after that date (Marshall, 2003; Nicolas and Bromwich, 2014a).

In addition to the characterization of the changes occurring in the 1970s, the SMB records will be one of the main sources for our reconstruction of polynya activity over the past centuries. The other data set is a synthesis of isotopic variations ($\delta^{18}$O) including 112 cores (Stenni et al., 2017a). SMB and $\delta^{18}$O are the two variables measured in ice cores selected here because they are used routinely to interpret past changes in precipitation and temperature over Antarctica. Furthermore, the syntheses available (Stenni et al., 2017a; Thomas et al., 2017) provide a reasonably good coverage over Antarctica, in particular in the South Atlantic sector where we expect the strongest signature of the great Weddell Sea polynya formation. Several model-data

comparisons have also been carried out using those variables (e.g., Klein et al., 2019; Dalaiden et al., 2020; Cavitte et al. 2020), which therefore provide a basis for the reconstructions using data assimilation proposed here.

Stenni et al. (2017a) and Thomas et al. (2017) selected only ice cores with a good time resolution and a low dating uncertainty. The dating error is thus small for the cores included in these syntheses with a maximum of a few years over the past centuries. This is essential for polynya detection since a sequence of opening may only last a few years. Here, we will be even more strict and select only a subset of those data with annual resolution and the lowest age uncertainty, following the choice of Medley and Thomas (2019) (see Table 1).

It may be difficult to make the distinction from available records between a year characterized by a few exceptional precipitation events such as atmospheric rivers that leave a large imprint on surface mass balance (e.g., Gorodetskaya et al., 2014; Turner et al., 2019) and the consequences of the opening of a polynya. As the focus here is on large open ocean polynyas that are assumed to occur in sequence of several years, a 3-year running mean is applied on the time series in the majority of our analysis. This provides a good balance between smoothing atmospheric events that may dominate at the interannual timescale while still being able to identify polynya sequences lasting a few years such as the one between 1974 and 1976. We also remove the trend over the period 1850-1992 in the ice core records. Those trends are likely due to a large extent to processes unrelated to polynya formation (Medley et al., 2018; Medley and Thomas, 2019). A part of the trend could also be due to a recent shift in polynya activity (e.g., de Lavergne et al., 2014) but it is impossible to disentangle the various contributions at this stage. We thus preferred to remove the trend to avoid misinterpretations. After detrending, we ensure that the mean of the ice core records before 1850 is the same as after 1850. This assumes a stationarity of the time series. Unfortunately, this procedure prevents us to compare the frequency of open ocean polynya formation during the 20$^{th}$ century with previous periods.

## 2.2 Model results

From a description of the observed temperature and precipitation changes in 1974, 1975 and 1976, it is impossible to disentangle the impact of polynya formation from the variability of the system that is not connected to the polynya itself. Observations are thus complemented by the results of model simulations. The first simulation was performed with the atmospheric model ECHAM5-wiso (Steiger et al., 2017; 2018). that has a spatial resolution of 1.125° and simulates explicitly the water isotopes. ECHAM5-wiso is driven by observed changes in sea surface temperature and sea ice concentration from the Met Office Hadley Centre's sea ice and sea surface temperature data set over the period 1871 to 2011 (Rayner et al., 2003 and updates). It provides thus a direct estimate of the model response in terms of temperature, SMB and δ$^{18}$O over Antarctica to observed changes in ocean surface conditions in 1974, 1975 and 1976. Before 1973, no satellite imagery is included in Rayner et al.'s reconstruction of the sea ice cover in the Southern Ocean and sea ice concentration is derived from Antarctic Atlas Climatologies. The model boundary condition is thus more uncertain that for the more recent period, with a direct impact on the ECHAM5-wiso results. In our analysis, we compared the conditions in 1974, 1975 and 1976 to a model climatology

established for the period 1958-2000 for consistency with the other data sources (the simulation results before 1958 are not used here). Nevertheless, our results in the region close to the Weddell Sea under the direct influence of the polynya are not sensitive to the reference period selected and are, for instance, very similar there if the reference period is shifted to 1979-2011 when satellite data is available and the uncertainties on the estimates of the ice concentration lower.

Several climate models display polynyas of various sizes and locations. Because of the triggering effect of Maud Rise, it is likely that, if large polynyas occurred in the Weddell Sea before the 1970s, they were also located close to the ones observed in the 1970s. Nevertheless, assessing the realism of simulated polynyas is difficult as we do not know if the sequence observed in the 1970s corresponds to the standard size of polynya we should expect in the current climate, if they were among the largest ones observed during the past centuries of if much bigger ones occurred earlier.

Here, two control simulations performed with the SPEAR (Seamless system for Prediction and Earth system Research) global climate model (Delworth et al., 2020), developed at the Geophysical Fluid Dynamics Laboratory are chosen because they display intermittent polynya formation whose size and characteristics share many elements with observed changes in the Southern Ocean (Zhang et al., 2019; Delworth et al., 2020). They have constant forcing corresponding to pre-industrial conditions and are not constrained by any observations. They could thus not reproduce the observed conditions in the 1970s but they simulate a large number of polynya events that allows a robust attribution of the impact of modeled polynya on the Antarctic continent. Furthermore, these simulation results provide the model prior in the data assimilation, as explained below. It is thus important to assess their characteristics compared to observations.

The two SPEAR simulations use the same ocean model, MOM6, with the SIS2 sea ice component (Adcroft et al., 2019), at a horizontal resolution of about 0.5° in the Southern Ocean. The first simulation, referred hereafter to as SPEAR_LO, includes the AM4 atmospheric component (Zhao et al., 2018) at a resolution of about 100 km. The second, SPEAR_AM2, uses AM2 (Anderson et al., 2004), an earlier version of the model at a resolution of about 200 km (see Zhang et al., 2020 for a longer description of the differences between the two simulations). For both simulations, we analyze here the last 1000 years of the experiments, corresponding to years 2000-3000 in SPEAR_AM2 and years 3000-4000 in SPEAR_LO.

### 2.3 Reconstruction methods

For reasons explained in Sect. 4, we have only selected the surface mass balance records for our reconstructions. The occurrence of polynyas over the past centuries is first estimated using a simple and direct method that will be described in the same section. Additionally, the history of past polynya formation is derived using data assimilation, a technique that has been applied recently to reconstruct climate fields, such as surface temperature and variables related to hydrology, over the past millennium (e.g., Goosse et al., 2012; Hakim et al., 2016; Franke et al., 2017; Steiger et al., 2018). Here, we apply the so-called offline or non-cycling data assimilation as in many previous studies (e.g., Hakim et al., 2016; Franke et al., 2017; Steiger et al., 2018; Klein et al., 2019), meaning that it is based on an existing ensemble and no additional simulation is performed in contrast to online data assimilation.

The objective of data assimilation as implemented here is to combine model results and observations to reconstruct the state of the climate system and to deduce from it the timing of occurrence of open ocean polynya in the Weddell Sea. The technique is based on a particle filter, following the implementation of Dubinkina and Goosse (2013) (see also Klein et al., 2019 and Dalaiden et al., 2020). An ensemble of simulated states is first obtained from annual means for al the year of the control runs of SPEAR_AM2 or SPEAR_LO. This forms what is referred to as the prior distribution. For every year of the reconstruction,

the likelihood of each of these simulated annual mean state is evaluated from the differences between the model results and observations. The goal is to obtain a quantitative estimate of the agreement between the observations for this specific year and each model state. From this likelihood, a weight is given to each model state. In other words, we evaluate the ability of each available model year to act as an analog for observed conditions during the selected year and give a higher weight to the best analogs. Formally, this provides the posterior distribution. More specifically, using those weights, we can compute a weighted

mean of the model states that should be, by construction, close to the observations used as constraint in the evaluation of the likelihood (the selected surface mass balance observations here). This weighted mean can also be calculated for all the other variables simulated by the model. Here, we focus on the changes in ocean convection as this can be used to construct an index for polynya formation as explained in Sect. 3.

   In our experiments with data assimilation, the records are averaged over ~500 x 500 km boxes, as coarse resolution models

are not expected to correctly represent smaller scales. These averages are shifted by a constant value and scaled to have the same mean and variance as the reconstruction of Medley and Thomas (2019) for the same boxes over the period 1941-1990. The surface mass balance has a large spatial variability at all spatial scales (Thomas et al., 2017; Laepple et al., 2019; Cavitte et al., 2020) and this procedure provides values that are not too sensitive to the mean conditions at specific locations. Technically, this can be considered in the present framework as part of the observation operator that allows the correspondence

between model space and observation space.

   A key element in data assimilation is to obtain a reliable estimate of the observation error. As classically done, we consider that the errors are not correlated and that the observation error covariance matrix is diagonal. Because of the large spatial variability mentioned above, we also assume here that the representation error is much larger that the measurement error (Thomas et al., 2017; Laepple et al., 2019; Cavitte et al., 2020; Badgeley et al., 2020) and we only include the contribution of

the former in our estimate of the error. The representation error is due in particular to the fact that the model is not able to simulate the small-scale processes that are included in the signal recorded in the archive (the so-called 'error of representation due to unresolved scales and processes', see for instance Janjić et al., 2018). The representation error is estimated by using a high resolution (approximately 27 km) simulation performed with the regional atmospheric model RACMO over the period 1979-2016 (van Wessem et al., 2018). It is obtained by calculating the standard deviation of the difference between the average

of the annual mean surface mass balance time series using only the RACMO grid boxes where ice core records are available and the true average over the continental part of the 500 by 500 km boxes. The mean of all the simulated series over the period 1976-2016 is removed before the standard deviation is calculated to focus on the time variability within the boxes, not on the

differences in mean snow accumulation. This standard deviation is then multiplied by a factor 0.6 to take into account the smoothing associated with the 3-year running mean applied to the time series (see above). We prefer this method over that

computing the standard deviation after applying a 3-year running mean on the time series of simulated results because of the small number of samples in the RACMO simulation.

## 3 Fingerprint of polynyas in observation and models

As discussed in the introduction, the polynya formation is expected to have a larger impact in winter but, as the resolution of the records is at best annual, we will focus here on annual mean values. The annual mean temperature in the reconstruction of

230 Nicolas and Bromwich (2014a) shows a large and clear positive anomaly over the years 1974-1976 in continental regions located close to the eastern Weddell Sea where the large polynyas appear in the 1970s. Specifically, higher temperatures are found between 50°W and the Greenwich Meridian, with a maximum reaching more than 2°C near the coast (Fig. 2a). The number of weather stations being low in Antarctica, this positive temperature anomaly is mainly influenced by observations at the Halley station, which is located at 75.6°S, 26.6°W and is the weather station that is the closest to the great Weddell Sea

polynya (Carsey, 1980). The other weather station with a long record in the region is Novolazarevskaya (70.8°S 11.8°E) which, by contrast, does not display particularly warm conditions at that time (Fig. 2b).

Halley station shows multidecadal variability with generally higher temperatures between 1970 and 1990 than in the following two decades. When applying a 3-year running mean to smooth interannual variability while keeping the signal associated with the observations of polynyas three year in a row, the maximum of the whole series is reached in 1976, i.e. during the polynya

formation period. However, the annual mean temperature anomaly compared to the period 1958-2000, with a value of 1.6°C, does not appear exceptional or out of the normal range of variability for the region (standard deviation of the annual mean temperature is 0.7°C). Furthermore, when taking a 3-year period, the maximum at Halley is not for the polynya years 1974-1976 but is shifted by one year (1975-1977). This may be due to the drift of the polynya toward the west from 1974 to 1976, moving closer to Halley Station in the final years (Carsey, 1980; Zwally et al., 1983). Furthermore, we analyze here annual

mean temperatures. The late freezing or the early melting of the sea ice in years preceding or following the polynya created large embayments but not strictly polynyas (Carsey, 1980; Zwally et al., 1983). The effect of the polynyas on annual mean temperatures can thus be extended in time even though no polynya strictly-speaking is formed in winter during those years.

Snow accumulation is also higher in 1974-1976 in the continental regions close to the polynya (Fig. 3a). The signal appears more spatially extensive in the SMB reconstruction of Medley and Thomas (2019) than for temperature in the Nicolas and

250 Bromwich (2014a) reconstruction, with positive values over nearly half of the continent. However, it is not clear if they are all related to the great Weddell Sea polynya formation.

The integrated surface mass balance over the continental region close to the polynya, defined here as the domain between 50W and 0°E northward of 80°S (see the sector indicated on Fig. 3a), has a local maximum in 1975 after applying a 3-year running mean (Fig. 3b). This corresponds exactly to the 3-year period with polynya formation but the maximum does not particularly

stand out in the time series and is, for instance, slightly lower than the one in 1995 when no major polynya was observed. In this region, the snow accumulation averaged over 1974-1976 is 12 Gt/y higher than the mean over the period 1958-2000. If a slightly larger domain covering 50°W-50°E is considered, a value of 24 Gt/y is obtained and 1975 is the absolute maximum over the period 1958-2000. Those numbers can be compared to a standard deviation of the SMB in those regions of 7 and 15Gt/y, respectively, and a standard deviation at the scale of Antarctica of 57 Gt/y over the period 1958-2000 in the Medley and Thomas (2019) reconstruction after applying a 3-year running mean.

The temperature signal in the ECHAM5-wiso simulation driven by observed sea surface temperature and ice concentration is very strong in the polynya region, with a warming reaching 2.5°C averaged over 1974-1976 (Fig. 4a). A weak warming is also seen over a relatively small coastal band between 50°W and 0°, with maximum values a bit lower than one degree in a few coastal regions. Those values on the continent are smaller than those observed at Halley station for the same time periods, suggesting either that the observed anomalies cannot be fully attributed to the polynya formation or that the simulations underestimate the temperature changes over the continent due to the great Weddell Sea polynya formation, or a combination of the two.

Precipitation in this simulation increases strongly over the polynya as well as over the continent between 50°W and 0° (Fig. 4b). However, precipitation tends to decrease between 0 and 50°E. Compared to observations, the signal appears thus more contrasted and the increase is only clear over the continent westward of the Greenwich meridian. At the daily timescale, the wind direction controls the location and amplitude of the temperature and precipitation signal related to polynya formation, as described in the model study of Weijer et al. (2017). An increase in precipitation over land is seen mainly when the winds comes from the north (in particular from the northeast, their figure 6) bringing the moist air originating in the polynya region, where evaporation is high, towards the continent. Close to the coast, the dominant easterly winds tend to push this moist air towards the west, explaining the larger signal found on the continent southwest of the polynya, as shown on Fig. 4. The net change in snow accumulation in the simulation averaged over 1974-1976 and integrated over the region 50°W-0 southward of 80°S is +13 Gt/y compared to the mean over the period 1958-2000 for the same region, a value surprisingly close to the one in the reconstruction of Medley and Thomas (2019).

The $\delta^{18}$O of precipitation is often related to temperature but the link can be weak and complex, in particular in coastal regions of Antarctica where the impact of polynyas is expected to be the largest (e.g., Masson-Delmotte et al., 2008; Sime et al., 2008; Holloway et al., 2016; Klein et al., 2019; Goursaud et al., 2019). This is even more problematic in the case of open-ocean polynya formation where large changes in the seasonality of precipitation are expected. Nevertheless, the pattern of annual mean $\delta^{18}$O of precipitation (Fig. 4c) associated with polynya formation is relatively similar to the one for temperature, with for instance high positive values over the polynya but only low positive ones over the continent except in a few regions close to the polynya. The ice cores record the signal in the precipitation accumulated over one year at least (e.g., Stenni et al., 2017a) and the $\delta^{18}$O of precipitation weighted by the amount of precipitation is a more adequate variable to compare to observed values. For this diagnostic (Fig. 4d), the signal becomes even weaker over the continent, with only a few coastal regions where the anomaly in annual mean $\delta^{18}$O reaches 0.5 ‰.

Consequently, the analyses of observations and model results indicate that, although the maximum of the surface response to
290 polynya formation is expected over the ocean in winter, the polynya opening has likely induced a warming over the continent in average over the period 1974-1976, as well as an increase in snow precipitation and a small modification of the $\delta^{18}$O of precipitation. However, the signal is not strong enough to detect without ambiguity the direct effects of the polynya compared to other processes that can also lead to large interannual climate variations. Unfortunately, for a polynya opening over 3 years only, we are unable to use any statistical test of significance that would give us a stronger conclusion.

In order to obtain complementary information on polynya dynamics and compare quantitatively the observed changes in the 1970s with the results of the SPEAR model, an index of open ocean convection and polynya activity is obtained in SPEAR_AM2 and SPEAR_LO control simulations by defining and calculating the annual mean mixed layer depth in the Eastern Weddell sea between 50°W and 50°E southward of 60°S. Open-ocean convection occurs only in winter but the variability of the mixed layer depth in the annual mean is controlled by the winter values. Therefore, using the annual mean
avoids making an arbitrary choice of which months to select for the average while convection can take place over a long period. Fig. 5 displays the regression of temperature and precipitation onto the above specified mixed-layer depth index for the last 1000 years of the SPEAR_AM2 and SPEAR_LO simulations.

For the last 1000 years of the simulation, SPEAR_AM2 simulates recurring polynyas with centers around 20°E that induce a warming of up to 0.5°C per standard deviation of the mixed-layer depth index over the continent close to the coast. Precipitation
over the continent also increases in response to polynya formation, with a total increase of snow accumulation over the region 50°W-0, northward of 80°S of 17Gt/y per standard deviation of the mixed layer depth index (30Gt/y per standard deviation for the region 50°W-50°E). This pattern is very similar to the one deduced both from the observations in the 1970s and the ECHAM5-wiso simulation.

In SPEAR_LO, the deep mixing and polynya formation in the Weddell Sea is concurrent with oceanic convection in the Ross
Sea (Zhang et al., 2020). Consequently, the warming and precipitation signal is more widespread over the continent. Close to the Greenwich meridian, the warming of the coastal regions is of the order of 0.5°C per standard deviation of the index as in SPEAR_AM2 simulation. The changes in snow accumulation over the region 50°W-0, northward of 80°S is 9 Gt/y per standard deviation of the index (41Gt/y per standard deviation for the region 50°W-50°E). Additionally, large warming and precipitation changes are observed in the simulation in the continental regions close to the Eastern Ross Sea and the Amundsen Sea.

**4 Reconstructing past polynya activity**

Standard methods applied to reconstructions covering the past millennium rely more or less directly on the correlation between the target (or predictand, here the polynya activity) and some predictors (for instance, selected ice core records) over the instrumental period in order to calibrate a statistical model (e.g., Mann et al., 2008; Jones et al., 2009; Christiansen and Ljungqvist, 2017; Stenni et al., 2017a). This statistical model is then applied over the full period were the predictors are
320 available to obtain a reconstruction of the target before the instrumental period. Here, this is not possible because the number

of samples during the instrumental period is too small for any successful calibration. We thus have to propose a slightly different approach.

We first apply a simple reconstruction method centered on the average of the records that are the most likely influenced by the formation of big polynyas in the Weddell Sea. The selection of these records is based on the information provided in Sect. 3.

A large warming has been measured at Halley weather station in 1974-1976. However, the magnitude and the spatial extent of the temperature increase that can be directly attributed to polynya formation is ambiguous in observations and models. No direct long term temperature record is available as input for our reconstructions. High resolution temperature estimates before the instrumental period are often derived from the $\delta^{18}O$ measured in ice cores but the $\delta^{18}O$ signal over 1974-1976 in the simulation results is not very clear and relatively large values are restricted to a small continental region (Fig. 4d). Furthermore,

when analyzing the available $\delta^{18}O$ observations in the region close to the Weddell Sea, no well-defined pattern is identifiable (Fig. 2a). The record closest to Halley station (Berkner Island-South ice core, see Table 1) even display a decrease in $\delta^{18}O$ over the time period 1974-1976 while, with a simplistic interpretation, we would have expected a higher value associated with the large warming observed in the region. Consequently, $\delta^{18}O$ records do not appear at this stage to be good candidates to reconstruct polynya activity with the simple methodology proposed here.

The reconstructions of the impact of the polynya formation on snow accumulation, based on ice cores, ECHAM5-wiso simulations for the 1970s and the control simulations with the SPEAR model display contrasted changes over the continent. Nevertheless, they all present an increase in the snow accumulation of similar magnitude between roughly 50°W and 0°. As open ocean polynya formation has been observed only during a few years, it is impossible to determine which element in the observed snow accumulation for this period is a response to polynya formation and which one is independent of the polynya.

Simulations with SPEAR models provide a sufficiently large number of events to unambiguously identify the signal associated with polynya formation in the models but this signal is influenced by model characteristics and biases. Nevertheless, the different sources of information are complementary and it seems reasonable to focus on the characteristics common to all those sources, i.e. the higher snow accumulation between 50°W and 0°. This does not imply that a response to polynya formation can only be seen in this sector, but we can likely identify the most robust signal there. Furthermore, this higher snow

accumulation has a clear and simple physical interpretation as stronger evaporation in the polynya region is expected to lead to more precipitation downwind (Weijer et al. 2017).

Consequently, we propose to base our reconstructions on surface mass balance records only. We focus on ice core records that are at least 150 year-long to avoid too many changes in the number of records over the period of analysis (Table 1). If we consider the broader region between the Antarctic Peninsula (which is clearly out of the domain of direct influence of the great

Weddell Sea polynya in our results) and 50°E, eight surface mass balance fit our criteria in the data selected by Medley and Thomas (2019) (Fig 3a). Two (Derwael Ice Rise IC12 at 26.34°E and H72 at 41.08°E) are located in regions where the polynya opening may induce a response but there is no consensus between the different estimates provided in Sect. 3. We thus do not keep them here. All the others are located at the margin of the region where the most robust changes have been identified (see

Fig. 3a, 4b and 5cd). We thus suggest to focus on these six records: Berkner Island (45.72°W), B31 DML-07 (3.34°W), B32-DML-05(0.01°W), B40 (0.07°E), Fimbulissen S100 (4.8°E) and B33-DML17 (6.5°E). The signal over 1974-1976 from those cores is not homogeneous (Fig. 3a) but this can be related to the influence of local processes and post-depositional alterations in the records that may lead to relatively large differences even between nearby sites (Thomas et al., 2017; Laepple et al., 2019; Cavitte et al., 2020). Nevertheless, four out of these six cores in the region have values in 1974-1976 higher than the mean over 1958-2000 (Fig. 3a), which is of course consistent with the spatial reconstruction of Medley and Thomas (2019) based on those records. As discussed in Sect. 2, all the time series are detrended over the period 1850-1992 and a 3-year running mean is applied prior to our analyses.

Since no direct calibration of polynya activity with instrumental observations is possible, we first standardize all the records by removing their mean and dividing by their standard deviation over the period 1941-1990. We then obtain the average of all those standardized records to obtain a qualitative index of polynya occurrence. This simple average has a physical justification:the common signal associated with polynya formation should be positive in all the records as polynya opening leads to more precipitation in the whole region of interest. The index is then scaled to have a value of 1 in 1975, to have an easy comparison with the observed polynya in the years 1974-1976. The methodology could be considered as the equivalent of the classical composite plus scale approach but with a slightly different final step compared to previous studies that performed the scaling to fit with the observed variance of the reconstructed variable (e.g., Mann et al., 2008; Jones et al., 2009; Christiansen and Ljungqvist, 2017; Stenni et al., 2017a).

The same records are used in the reconstructions with data assimilation. In theory, data assimilation should be able to handle observations outside of the region where the signal is the strongest. However, extracting the information on polynya activity potentially included in those records requires that the model simulates well the covariance between the regions where the polynya forms and the ones where those records are available. As it is difficult to evaluate the model performance on this aspect, we have decided to focus on the six ice core records that are the most directly and strongly influenced by the polynya opening, like for the statistical method. As in Sect. 3, the index of polynya activity is computed from the annual mean mixed layer depth in the Eastern Weddell Sea between 50°W and 50°E. The index is scaled to have a value of 1 in 1975 and the average over the period 1941-1990 has been removed to be consistent with the simple statistical reconstruction.

Because of the small number of records and the difficulty to perform an independent validation, a formal investigation of the uncertainties of our reconstructions is out of the scope of this study. Nevertheless, the robustness of our results can be estimated by comparing the indices obtained in the three methods: the statistical reconstruction and the two reconstructions with data assimilation using SPEAR_AM2 and SPEAR_LO simulations as priors, respectively (Fig. 6a). The two SPEAR simulations, which each display a polynya opening in a slightly different location in the Weddell Sea and an impact of these polynyas on snow accumulation over land characterized by a different pattern (Fig. 5), provide a rough range of the uncertainties associated with the simulation selected for the data assimilation. The sensitivity of the reconstruction to the selection of the records is also estimated by performing, for each method, alternative reconstructions based on all the combinations of five out of the six ice core records (i.e., we excluded each record one-by-one). The uncertainty for each method is then obtained from the standard

deviation of the seven reconstructions (the standard reconstruction including the six records and the six reconstructions with five records) (Fig. 6 bcd).

The three reconstructions based on the three methods display significant differences during some periods, such as the 1950's, (Fig. 6a) and the choice of the records has a large impact (Fig. 6bcd). This illustrates the large uncertainties still present at this stage in our estimates of past changes in open-ocean polynya formation. Nevertheless, for the last 150 years, the reconstruction generally agree on the timing of the largest peaks, likely corresponding to polynya formation. In particular, all the times series show a maximum in 1975, corresponding well to the period 1974-1976 with the 3 year running mean applied. This implies

that we are able to robustly reproduce the opening of the great Weddell Sea polynya. Smaller peaks are also observed in some reconstructions, for instance in 1983 for the statistical reconstruction (index of 0.65), while these years are not considered as particularly prone for the formation of smaller polynyas (Campbell et al., 2019). The index seems thus able to identify the known period with large open ocean polynyas but may have troubles to discriminate them clearly from years with high snow accumulation in the sector that may simply be caused by specific atmospheric conditions. The persistence of the polynyas over

a few years helps to reduce the noise due to random atmospheric processes but this is likely not enough. It is reasonable to consider that large open ocean polynyas should systematically lead to a widespread anomaly in the surface mass balance in the continental region that is downwind from the polynya and thus a high value of the index. By contrast, a high snow accumulation is not necessarily caused by a polynya. In this framework, we can thus make the hypothesis that the index provides more 'false positive' for polynya events than events we completely miss.

Over the period 1850-1990, a local maximum is found in the three reconstructions in 1882, with a value larger than in 1975 for the statistical reconstruction. Relatively high values of the index are also found in 1934 in the three reconstructions as well as some other peaks in individual reconstructions. Gordon (1982) made the hypothesis that a polynya occurred just before an oceanic cruise performed in 1962. The years before those observations are characterized by a prolonged period with a relatively high index in the statistical reconstruction but no value above 0.7 in any of our reconstruction. Such values may correspond to

changes in snow accumulation due to polynyas smaller than the one observed in 1975, but this is impossible for us to determine if the explanation is valid from the available records. Additionally, from an analysis of early satellite imagery, Meier et al. (2013) made the hypothesis that a polynya formed in the Weddell Sea in 1964, although it is difficult to make the distinction, based on the few available satellite images between polynya opening and the presence of leads and clouds in the region. 1964 is characterized by a low value of the index in all our reconstructions. This does not at all rule out the formation of a short-

lived event in 1964 but suggests that no equivalent to the polynya formation observed in 1974-1976 occurred in this period. For the pre-industrial period, high values of the index are found regularly in nearly every century (Fig. 7). To be more quantitative, two threshold values for the index have been applied to detect polynya formation on Figs. 8 and 9. The value of 1 corresponds to events that have a similar imprint as the great Weddell polynya while events with a smaller impact over the continent can still be detected with a value of 0.8. In each of our reconstruction, a year with an index higher than this value of

0.8 should still correspond to large events but the risk that a year with an index higher than 0.8 does not correspond to an open ocean polynyas is higher (the 'false positive' mentioned above) than for a threshold of 1.

The open ocean polynyas appear mainly as isolated events lasting a few years only, as observed in the 1970s. In addition to the isolated events, more persistent sequences (although not continuous) are also reconstructed in particular over the periods 1350-1400 and 1600-1650 (Figs. 8 and 9). By contrast, no polynya is reconstructed during some other periods such as the years 1500-1550 in all the reconstructions and for both thresholds. This may be the signature of centennial-scale variability in polynya activity. Nevertheless, as we go back in time, the absence of event can simply be due to the dating uncertainties. A shift by a few years only between the records can lead to an event being attributed to different years in the different time series and thus to a muted value of the index when they are averaged. Low frequency variations of the surface mass balance in the sector, due to processes independent of polynya formation, could also modify the background state over which the polynya signal is imprinted. A higher mean snow accumulation would then lead to a higher chance to pass the threshold of 0.8 or 1 while a lower mean snow accumulation would imply that only the big polynyas would be detected. Finally, because of the detrending applied to the time series, we cannot compare the polynya activity over the past century with earlier periods.

## 5 Discussion and conclusions

Large and persistent open ocean polynyas have a major impact on the ocean surface at high latitudes, on ocean dynamics and on the deep ocean properties, as highlighted in many studies. Their imprint on the Antarctic continent has been much less investigated. Because of the small number of events, disentangling precisely the signal at the ice sheet surface coming from an open ocean polynya from other elements of the climate variability is impossible using the instrumental data only. Nevertheless, instrumental data and surface mass balance reconstructions suggest a clear impact of the great Weddell Sea polynya in 1974-1976 on the continent, at least in the sector between roughly 50°W and 0°E. A comparison of the observed changes with the results of an atmospheric model driven by observed sea surface temperature and sea ice concentration suggests an annual mean warming of less than one degree in coastal regions and an additional snow accumulation averaged over the sector of about 10 Gt/year during the polynya formation compared to average conditions.

Because of this impact of the open ocean polynyas on the Antarctic ice sheet, it is tempting to use ice core records to reconstruct the occurrence of polynyas before the instrumental period. Surface mass balance records are the best candidates for an initial reconstruction because of their availability in the region close to the Weddell Sea. Furthermore, from robust physical arguments, polynya formation should be associated with an increase in snow accumulation in the sector downwind of the polynya. The signal is present in temperature as well but is weak for the isotopic composition of the snow, which is often considered as a proxy for temperature but whose interpretation is complex here in particular because of potential changes in the seasonality of precipitation due to open ocean polynya formation.

We have thus reconstructed an index of polynya activity based on the surface mass balance records, using a simple average of the standardized series available in the sector 50°W and 5°E as well as data assimilation constrained by those records over the period 1250-1990. This reconstruction remains qualitative at present and the uncertainties are still large. The surface mass balance changes caused by polynya formation are not exceptional enough to distinguish the origin of a large value without

ambiguity. Furthermore, low frequency variations in surface mass balance not related to polynya activity and dating

uncertainties in the records, which could potentially have a very large impact for events lasting at most a few years, also complicate the detection of polynyas.

Additional information is necessary to identify the years with a high value of our index of polynya activity but that actually do not correspond to a polynya. First, additional surface mass balance records would allow a reduction of the uncertainties and to extract more clearly the signal coming from the polynya, in particular if they are obtained in low elevation regions in the sector

10°W-30°W (see for instance Goel. et al. 2020 for potential drilling locations), and where all models selected here simulate the largest response to the formation of open-ocean polynya in the Weddell Sea. Furthermore, several proxies based on the chemical composition of Antarctic ice cores have been related to changes in sea ice concentrations (e.g., Abram et al., 2013; de Vernal et al., 2013; Thomas et al. 2019). For open ocean polynya, sea salt aerosols provide interesting perspectives as their source at the surface of polar oceans strongly depends on the presence of sea ice (e.g., Levine et al., 2014; Rhodes et al., 2018).

The formation of an open ocean polynya in winter should thus have a large impact on the sea salt input to the atmosphere in the Weddell Sea. This signal could then be transported to the continent by the winds and recorded in ice cores. Although biological activity in the polynya itself in winter may be limited, the formation of the open ocean polynya can have an impact on the vertical structure of the water column, on light availability, and nutrient input to the surface layer and thus on biological production later in the season (e.g., von Berg et al., 2020). This might also be recorded in ice cores, for instance in their

methanesulfonic acid content. These proxies have been used to reconstruct coastal polynyas (e.g., Rhodes et al., 2009; Criscitiello et al., 2013; Mezgec et al., 2018). The specific impact of a large open ocean polynya on these paleoclimate records is not well known but some of these records have likely a signal related to open ocean polynya formation that is large enough to place additional constraints on the reconstructions and reduce our uncertainties. Finally, if a high-resolution ocean sediment core could be collected at a location sensitive to the open ocean polynya formation in the Weddell Sea, this would provide

complementary information that could be combined with the ice core records to refine the reconstructions, in particular to constrain the low frequency variations of polynya activity.

In parallel, to those observations close to the region of polynya formation, another constraint on our reconstructions could come from a comparison to estimates of past large-scale changes that are expected to favor open ocean polynya formation or that would be a consequence of their occurrence. In particular, changes in surface winds influence the horizontal oceanic

circulation, potentially inducing upwelling of deep waters and thus a salt input in the surface layers creating conditions more prone to deep convection (Cheon et al., 2015; Campbell et al., 2019; Kaufman et al., 2020). In this framework, it has been argued that a persistent negative phase of the Southern Annular Mode (which is the main mode of atmospheric variability in the Southern Hemisphere extra-tropics) in the preceding decade could have created favorable conditions for the formation of the Weddell polynya in the 1970s (Gordon et al., 2007; Kaufman et al., 2020). Reconstructions of SAM over the past millennia

(Abram et al., 2014; Dätwyler et al. 2019) indicate generally low values between 1350 and 1700. This could be consistent with the high occurrence of polynya formation reconstructed from some periods such as 1350-1400 and 1600-1650 but the uncertainties are too large to be able to reach strong conclusions on the agreement with our reconstructions.

The great Weddell Sea polynya of the 1970s had a large impact on the ocean state. The heat loss has been estimated to be of the order of 0.4 $10^{21}$ J year$^{-1}$ (Gordon, 1982). This corresponds to 4% of the heat stored in the ocean in response to human-induced perturbations in recent years (about 1 $10^{22}$ J year$^{-1}$, Lyman and Johnson, 2014; Resplandy et al., 2018). Furthermore, the deep convection occurring in polynyas can lead to the formation of Antarctic Bottom Water (AABW). AABW is a key water mass, present over the majority of the ocean floor, and occupies more than 30 % of the global ocean volume (Mantyla and Reid, 1983; Johnson, 2008). It has its main origin at the margin of the Antarctic continent when cold and salty waters, formed on the Antarctic continental shelves because of brine rejection during sea ice formation, sink to great depths (Foster and Carmack, 1976; Purkey et al., 2018). During the polynya years in the 1970s, the volume of surface water entrained at depth in the Weddell Sea was higher than along the continental margin (Gordon, 1982) but the overall contribution of deep convection in AABW formation is highly uncertain (Martinson et al., 1981; Gordon 1982). Nevertheless, frequent polynya formation during some periods should lead to a higher production of AABW in the Weddell Sea and potentially to a modification of its characteristics. It has been speculated that AABW formation because of open ocean convection was higher during the period 1350-1850, potentially because of colder conditions at that time, than in the more recent past (Broecker et al. 1999). By contrast, from the analyses of marine records on the southern Chilean margin in the SE Pacific, Collins et al. (2019) suggest that AABW formation was weaker after 1400 CE compared to the periods before. Our reconstruction does not suggest such a systematic, long-term shift but the larger number of polynyas recorded for some periods should have consequences on deep waters that can be detected away from the Weddell Sea.

Our reconstructions provide only a preliminary step whose goal is to stimulate more investigation on the subject. Our target is also limited to polynyas similar to the great Weddell polynya of the 1970s. This implies that we have not addressed the occurrence of smaller polynyas, that may be more frequent, or of open ocean polynyas with a signature different from the Weddell polynya of the 1970s, in particular those that may have been present in other sectors of the Southern Ocean.

Despite those limitations, we are still able to reach some conclusions about the frequency of polynya formation. First, the polynyas like the one observed in 1974-1976 are not frequent in the past millennium, occurring only a few times per century at most. The mean number of years with open ocean polynya for the whole period ranges from 1.8 to 3.7 years per century in our three reconstructions (criteria at 0.8 with a value of one corresponding to the great Weddell polynya of the 1970's). Second, the sequences of polynya opening tend to last a few years only, with no clear periodicity. Some exceptions may have occurred with high surface mass balance values potentially associated with formation of polynyas during several decades. Nevertheless, as we go back in time, the uncertainties on the potential processes controlling the precipitation are larger, as are the dating uncertainties. Third, on a more technical point of view, the observed changes associated with polynya formation are similar to the ones given by global climate models that display realistic open ocean polynyas in the Weddell Sea, indicating that those models can be used for data assimilation. This technique is thus very promising for future reconstructions of polynya activity.

**Data availability**. Instrumental observations can be obtained on the READER site (https://legacy.bas.ac.uk/met/READER/, last access: 9 April 2020). The reconstruction of Antarctic surface air temperatures based on instrumental data is available at

http://polarmet.osu.edu/datasets/Antarctic_recon/ (last access: 4 July 2018, Nicolas and Bromwich, 2014b). $\delta^{18}$O time series are available at the NOAA World Data Center for Paleoclimatology (https://www.ncdc.noaa.gov/paleo-search/study/22589, last access: 18 March 2018, Stenni et al., 2017b). The surface mass balance time series comes from https://data.bas.ac.uk/full-record.php?id=GB/NERC/BAS/PDC/00940 and the reconstruction of Medley and Thomas (2019) is available at https://earth.gsfc.nasa.gov/cryo/data/antarctic-accumulation-reconstructions. The results of the ECHAM5-wiso simulation covering the 1871–2011 period can be downloaded from https://doi.org/10.5281/zenodo.1249604 (Steiger, 2018). RACMO2 data are available by request to Jan Lenaerts (jan.lenaerts@Colorado.EDU). The SPEAR-LO and SPEAR-AM2 results are available by request to Liping Zhang (liping.zhang@noaa.gov). The reconstructions of polynya activity are available at https://zenodo.org/record/1319334#.W2go2Lg6-M; last access: 13 November 2020) (Goosse et al., 2020).

**Author contributions**. HG designed the study. QD, MC and HG performed the analysis and made the figures. LZ provided SPEAR model results and advice on their use. All the authors contributed to the discussion and interpretation of the results. HG led the writing of the manuscript with contributions from all authors.

**Competing interests**. The authors declare that there is no conflict of interest.

**Acknowledgements**. This work was supported by the Belgian Research Action through Interdisciplinary Networks (BRAIN.be) from Belgian Science Policy Office in the framework of the project "East Antarctic surface mass balance in the Anthropocene: observations and multiscale modelling (Mass2Ant)" (contract no. BR/165/A2/Mass2Ant). Hugues Goosse is research director within the F.R.S.-FNRS. Quentin Dalaiden is a research fellow with the Fonds pour la formation á la Recherche dans l'Industrie et dans l'Agronomie (FRIA-Belgium). L. Zhang is supported through UCAR under block funding from NOAA/GFDL. We would like to thank all the scientists that collected and share their ice core records as well Nathan Steiger for sharing the output of the ECHAM5-wiso simulation, Jan Lenaerts for sharing the output of the RACMO simulation and Rachael Rhodes, Jean-Louis Tison and François Fripiat for stimulating discussions on polynya reconstructions.

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

**Table 1 –Ice-core records used in this study (based on Medley and Thomas, 2019 and Stenni et al., 2017). The surface mass balance from the first 6 records are used in the reconstructions of the polynya activity. The other records (*in italic*) are only displayed on Figs. 2 and 3.**

| Number | Site name | Longitude (°) | Latitude (°) | Altitude (m) | Years CE | Reference |
|---|---|---|---|---|---|---|
| 1 | Berkner Island (South) | -45.72 | -79.57 | 890 | 1000–1992 | Mulvaney et al. (2002) |
| 2 | B31Site DML07 | -3.43 | -75.58 | 2680 | 1000–1994 | Graf W. et al. (2002) |
| 3 | B32Site DML05 | -0.01 | -75.00 | 2892 | 1248–1996 | Graf W. et al. (2002) Sommer et al. (2000) Oerter et al. (2000) |
| 4 | B40 | 0.07 | -75.00 | 2892 | 1–2010 | Medley et al. (2018) |
| 5 | Fimbulisen S100 | 4.8 | -70.24 | 48 | 1737–1999 | Kaczmarska et al. (2004) |
| 6 | B33Site DML17 | 6.5 | -75.17 | 3160 | 1250–1997 | Graf W. et al. (2002) Sommer et al. (2000) Oerter et al. (2000) |
| *7* | *Derwael Ice Rise IC12* | *26.34* | *-70.25* | *450* | *1744–2011* | *Phillipe et al. (2016)* |
| *8* | *H72* | *41.08* | *-69.2* | *1214* | *1832-1999* | *Nishio et al. (2002)* |
| *9* | *IND 22B4* | *11.54* | *-70.86* | *500* | *1533-1994* | *Laluraj et al. (2011)* |
| *11* | *NUS 08-7* | *1.6* | *-74.12* | *2673* | *1382-2008* | *Steig et al. (2013)* |
| *12* | *NUS 07-1* | *7.94* | *-73.72* | *3174* | *1706-2005* | *Steig et al. (2013)* |

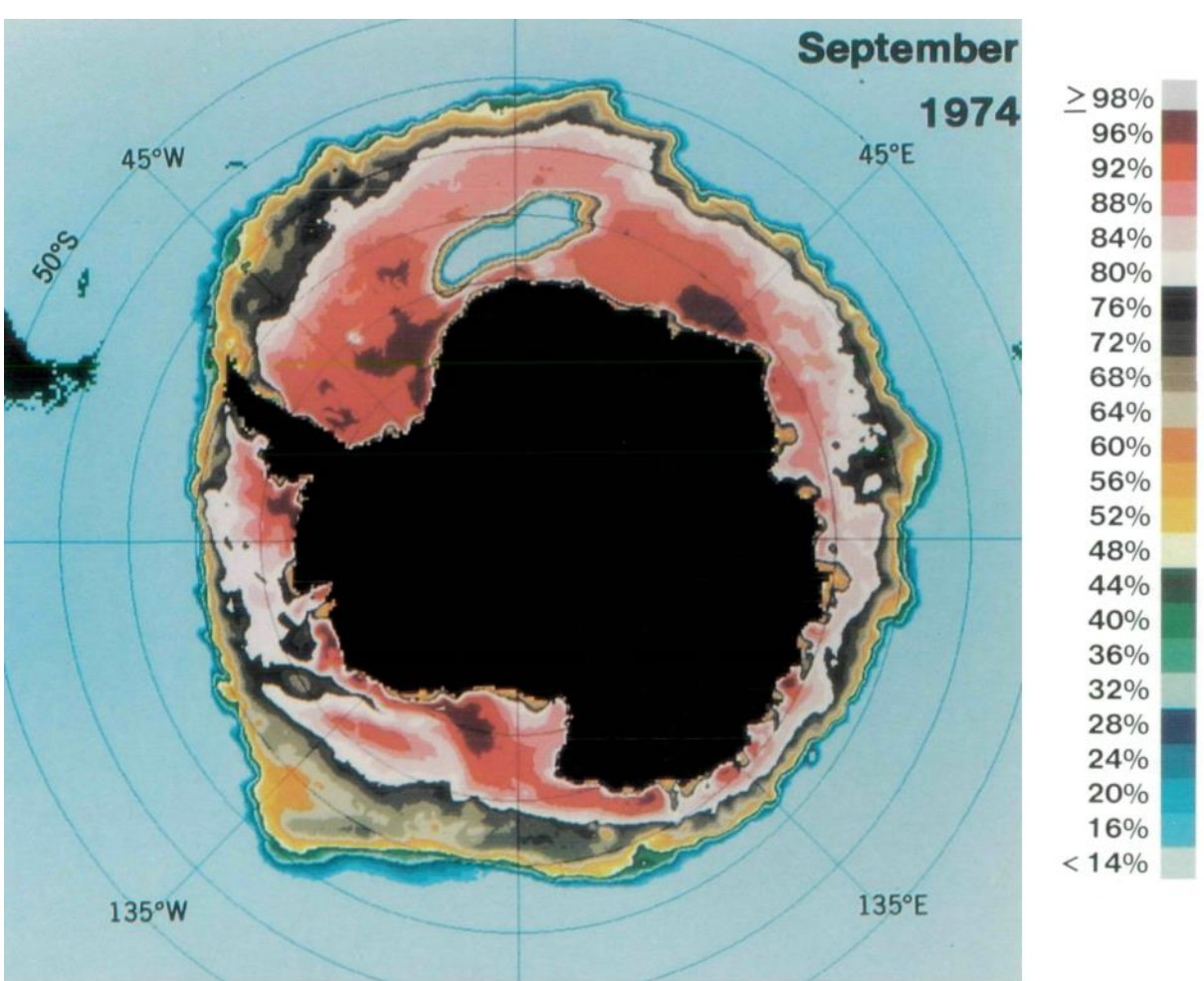

**Figure 1: The Weddell Sea polynya in Austral winter, September 1974. Violet and red correspond to a high sea ice concentration and light blue to open ocean. The great Weddell Sea polynya is visible across the Greenwich meridian. Figure from Zwally et al. (1983).**

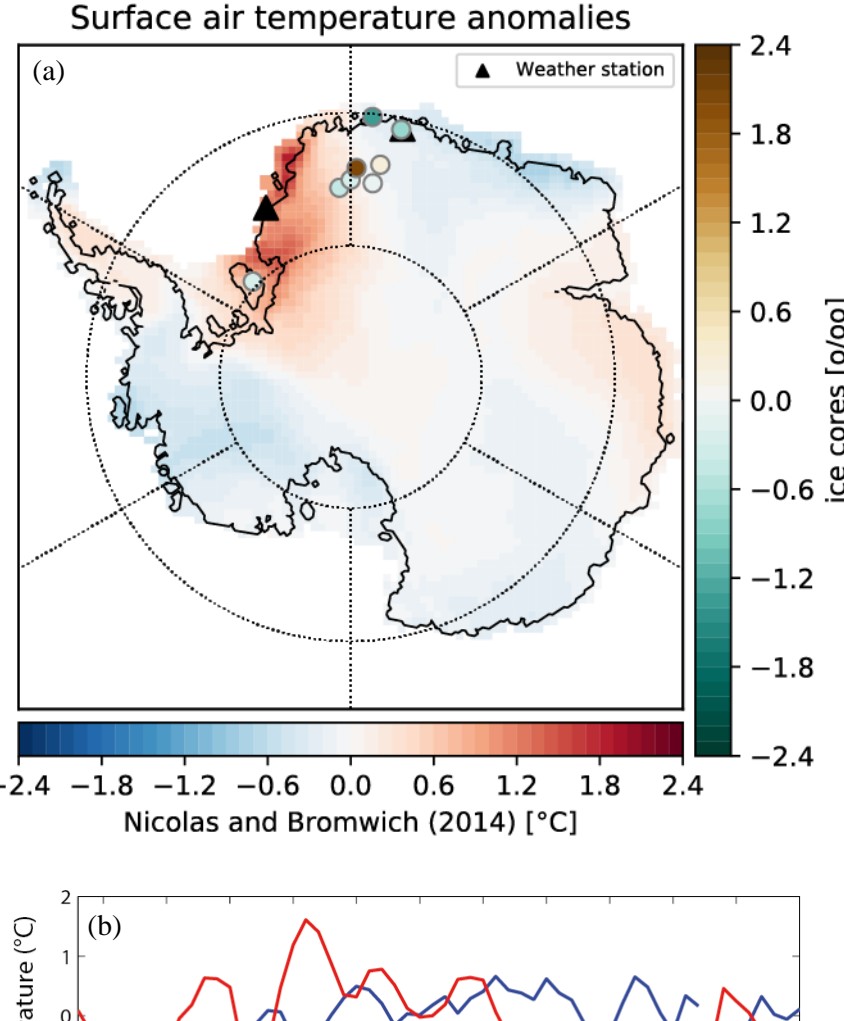

**Figure 2: a) Annual mean temperature anomaly (°C) averaged over 1974-1976 compared to the period 1958-2000 in the Nicolas and Bromwich (2014a) reconstruction. The dots correspond to the δ¹⁸O (‰) anomalies for the same period in ice cores in the region of interest. b) Annual mean temperature anomaly (°C) at Halley (red) and Novolazarevskaya (blue) weather stations (highlighted in panel a). A 3-year running mean has been applied to the series.**

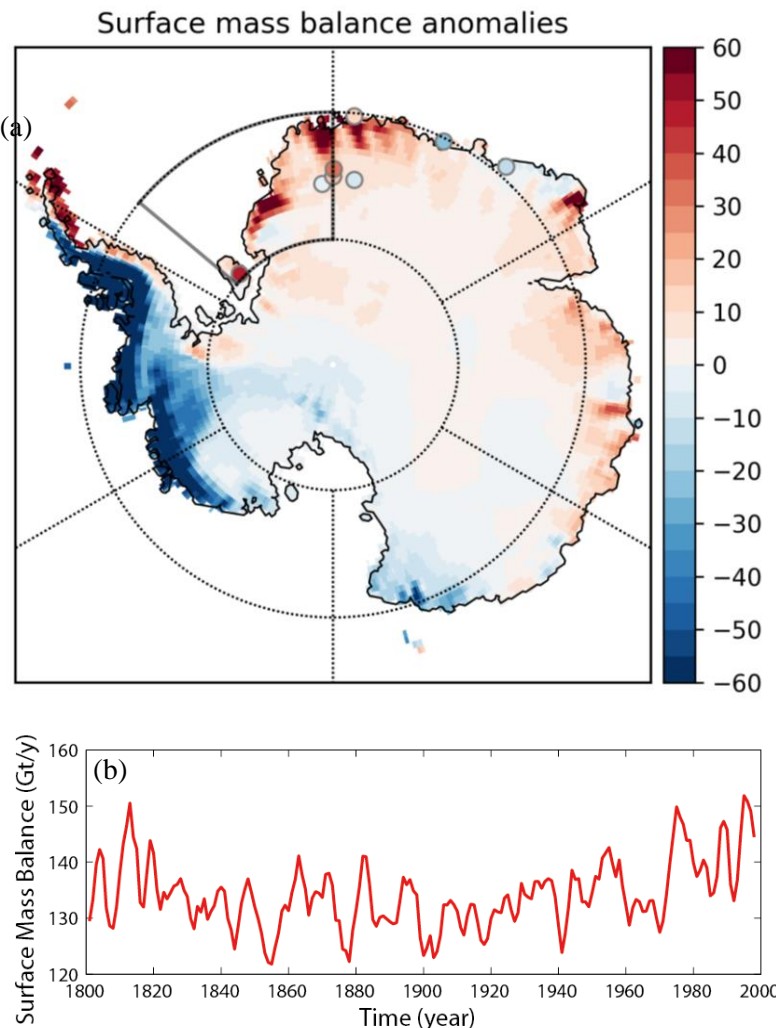

**Figure 3: a) Anomaly of SMB (mm w.e. /y) averaged over 1974-1976 compared to the period 1958-2000 in the Medley and Thomas (2019) reconstruction. The dots correspond to the estimates from the ice cores in Table 1. On the figure, the B40 core has been shifted northward by 0.6° in order to avoid too large an overlap with nearby cores. b) SMB (Gt/y) integrated over the grounded ice sheet between 50 W 0°E, north of 80°S (box on panel a) in Medley and Thomas (2019) reconstruction. A 3-year running mean has been applied to the series.**

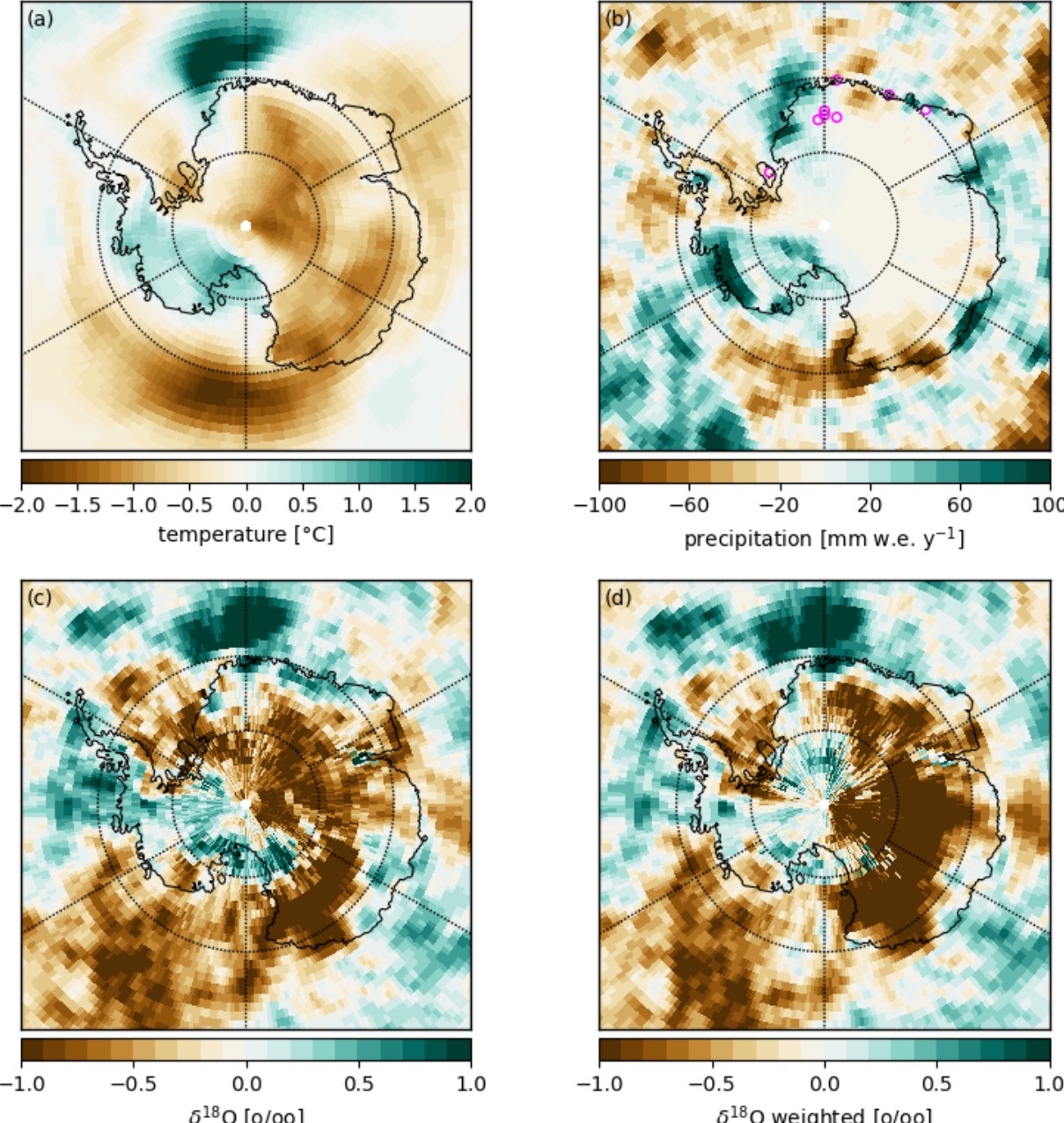

**Figure 4: Anomaly of (a) annual mean temperature (°C), (b) precipitation (mm w.e./y), (c) mean δ18O of precipitation (per mil) and (d) mean δ18O weighted by the precipitation amount (per mil) averaged over 1974-1976 compared to the period 1958-2000 in a simulation performed with ECHAM5-wiso. The circles on panel b correspond to location of the ice cores in Table 1 with the B40 core shifted northward by 0.6° in order to avoid too large an overlap with nearby cores.**

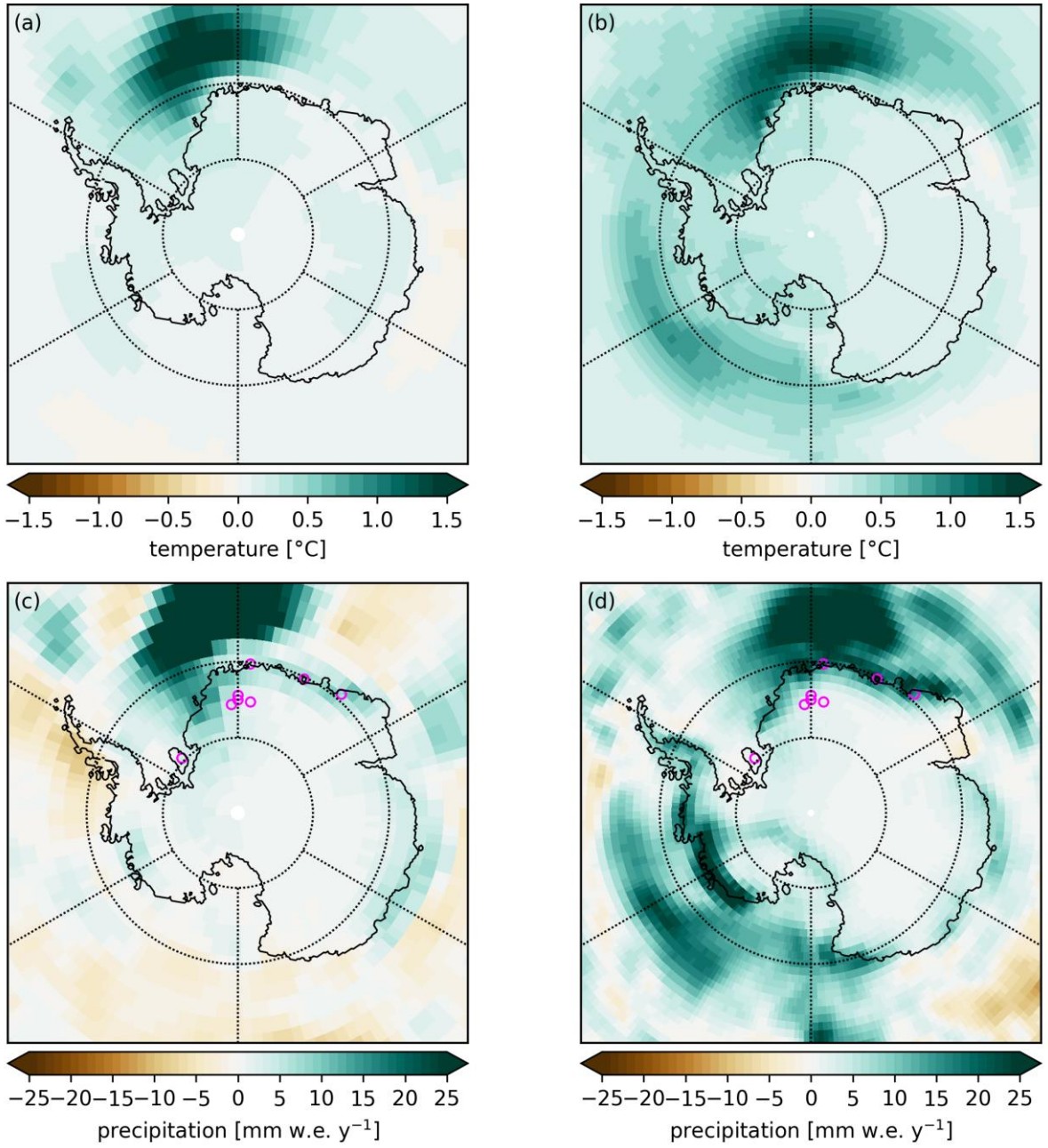

**Figure 5: (a)** Regression of annual mean temperature (°C) and **(c)** precipitation (mm w.e. /y) scaled to correspond to one standard deviation change of the annual mean ocean mixed layer depth in the Eastern Weddell Sea between 50°W and 50°E, southward of 60°S over the years 2000-3000 of the SPEAR_AM2 simulation. Same in **(b, d)** for the years3000-4000 of the SPEAR_LO simulation. The circles on panels c and d correspond to the location of the ice cores in Table 1 with the B40 core shifted northward by 0.6° in order to avoid too large an overlap with nearby cores.

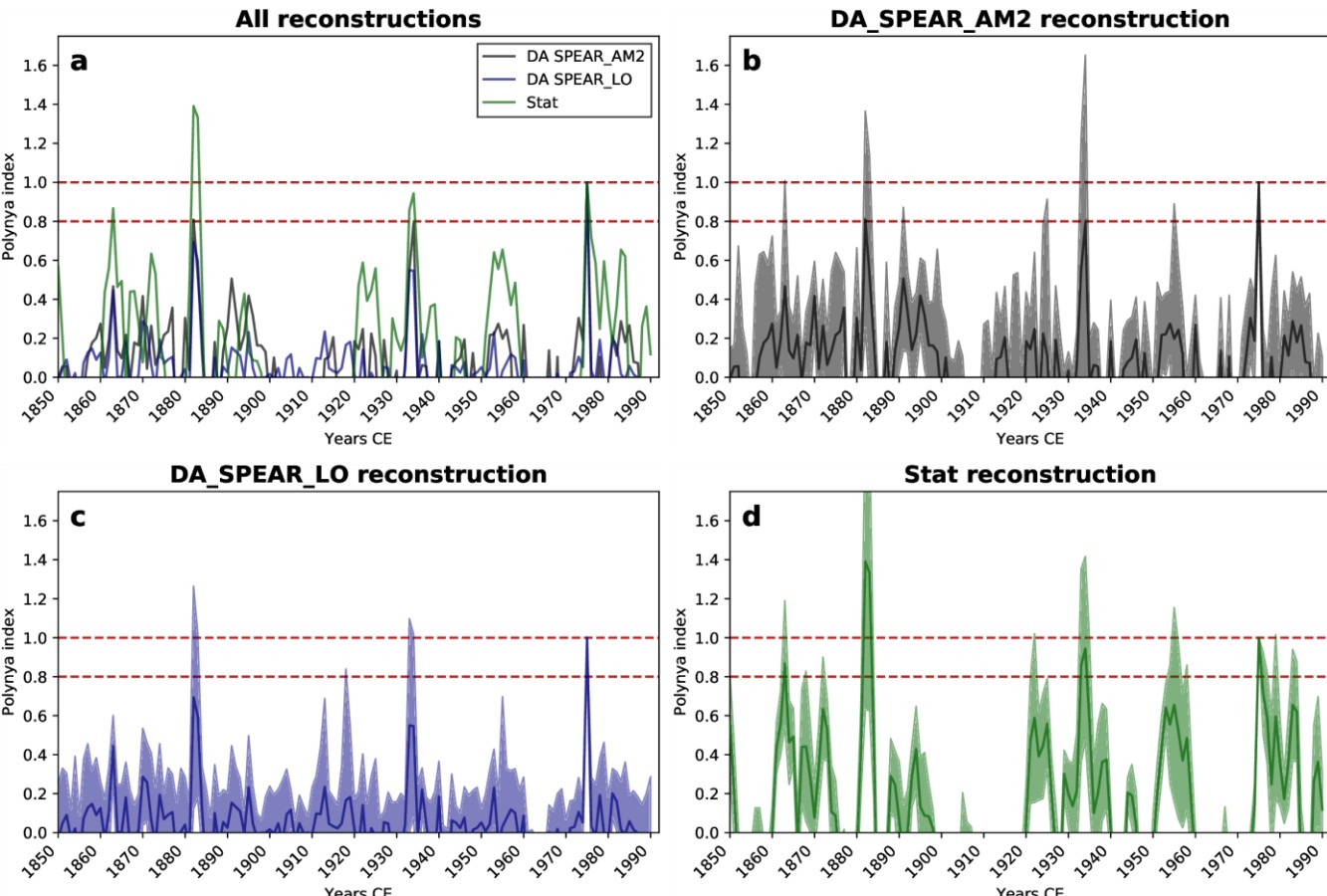

**Figure 6: a) Index of polynya activity for the years 1850-1990 based on 6 surface mass balance records using data assimilation with SPEAR_AM2 (DA SPEAR_AM2 black), using data assimilation with SPEAR_LO (DA SPEAR_LO, blue) and a simple average of standardized time series (Stat, green). Index of polynya activity with the uncertainties estimated from the standard deviation of the seven reconstructions using six and all the combinations of five different records for the reconstruction b) using data assimilation with SPEAR_AM2 c) using data assimilation with SPEAR_LO and d) a simple average of standardized time series.**

885

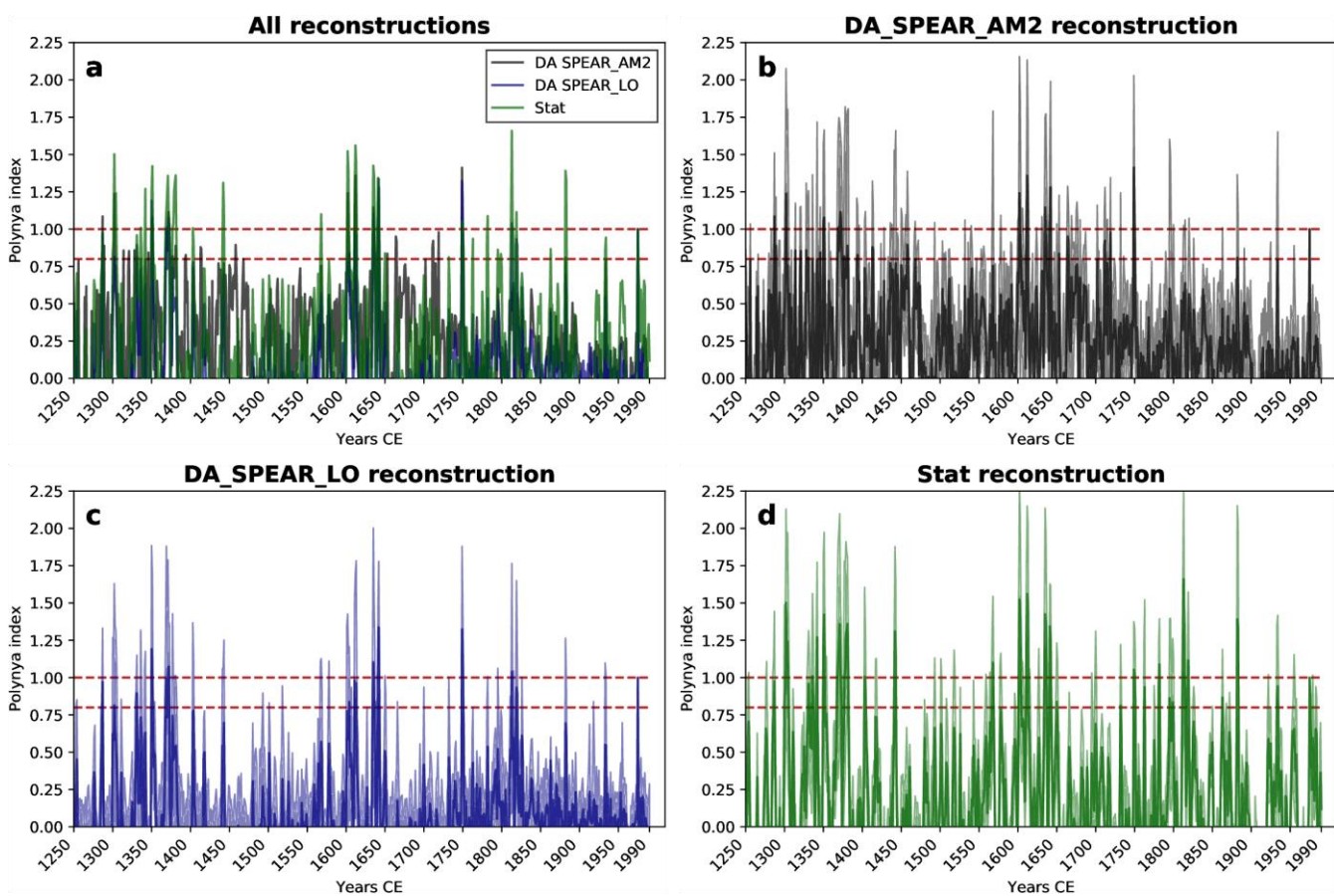

Figure 7: Same as Fig. 6 for the years 1250-1990.

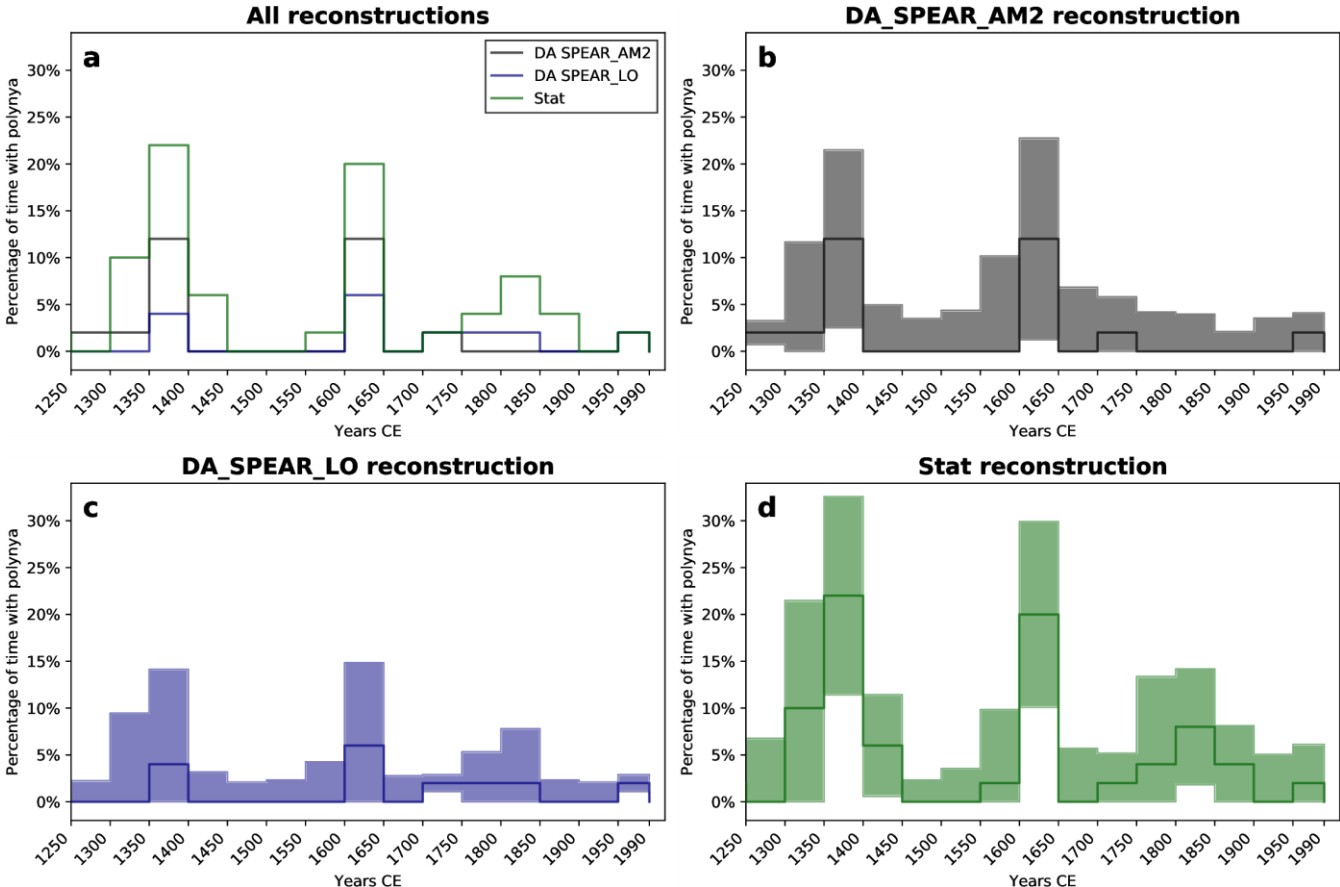

**Figure 8: a)** Percentage of the years with the index of polynya activity higher than 1 per 50 year time interval in the reconstructions using data assimilation with SPEAR_AM2 (DA SPEAR_AM2 black), using data assimilation with SPEAR_LO (DA SPEAR_LO, blue) and a simple average of standardized time series (Stat, green). Percentage of the years with the index of polynya activity higher than 1 with the uncertainties estimated from the standard deviation of the seven reconstructions using six and all the combinations of five different records for the reconstruction **b)** using data assimilation with SPEAR_AM2 **c)** using data assimilation with SPEAR_LO and **d)** a simple average of standardized time series.

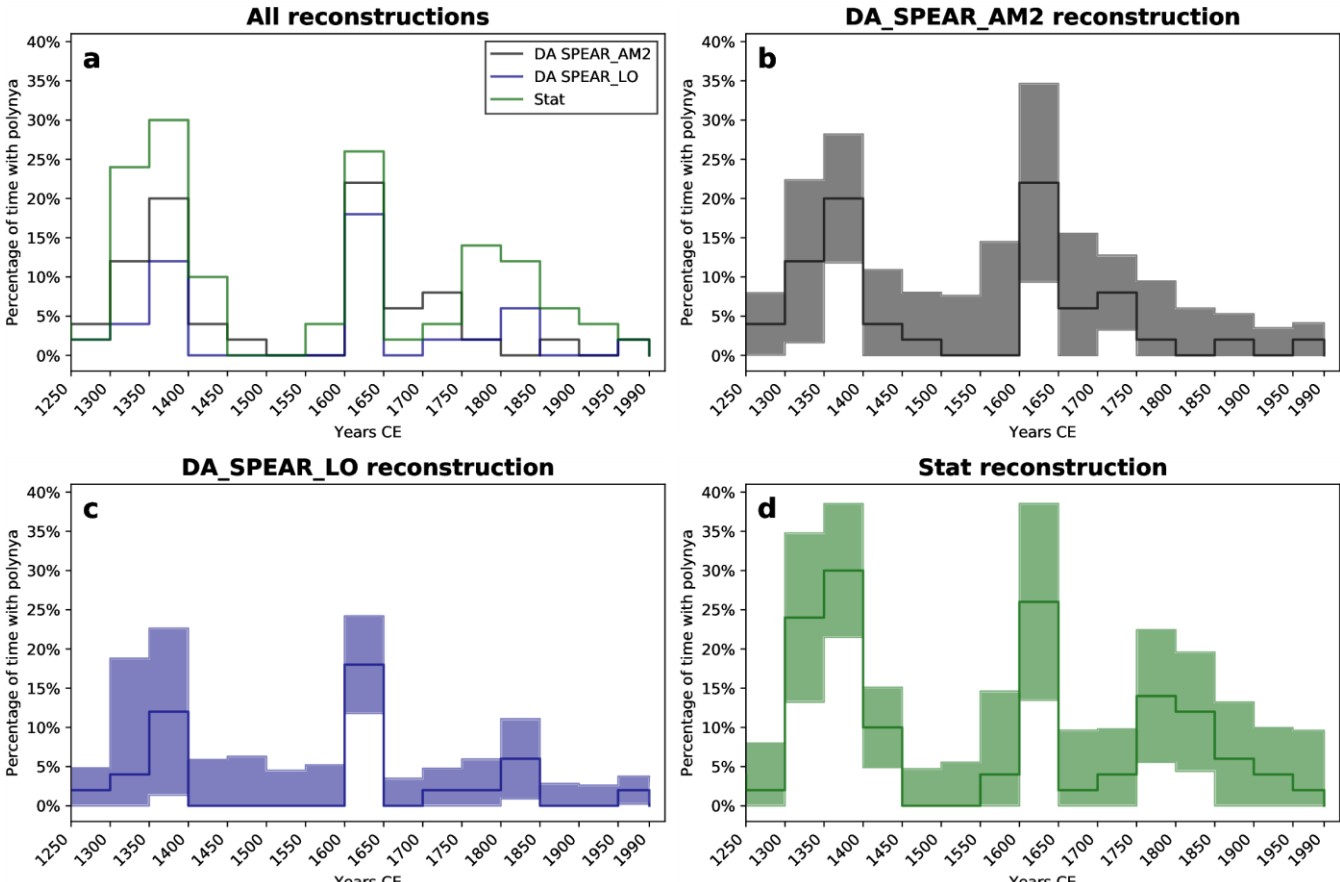

**Figure 9: Percentage of the years with the index of polynya activity higher than 0.8 (b) per 50 year time interval in the reconstructions using data assimilation with SPEAR_AM2 (DA SPEAR_AM2 black), using data assimilation with SPEAR_LO (DA SPEAR_LO, blue) and a simple average of standardized time series (Stat, green). Percentage of the years with the index of polynya activity higher than 0.8 with the uncertainties estimated from the standard deviation of the seven reconstructions using six and all the combinations of five different records for the reconstruction b) using data assimilation with SPEAR_AM2 c) using data assimilation with SPEAR_LO and d) a simple average of standardized time series.**