# Peer review of "Can we reconstruct the formation of large open ocean polynyas in the Southern Ocean using ice core records?"

_Climate of the Past, 2020_

## Short Comment (SC1) · 7 Sep 2020

I want to thank the authors for submitting this fascinating work to Climate of the Past. The journal allows members of the broader scientific community to comment on preprints during the open discussion period. I thought I'd use the opportunity to provide a few brief thoughts, which are not intended to be an exhaustive review.

As someone with a close interest in the Weddell polynya phenomenon (e.g. Campbell et al. 2019), I found this study to be particularly exciting. The techniques used by the authors to identify the fingerprints of past polynyas in ice core records of surface mass balance (among other data) seem well-motivated and promising. Without commenting

further on the technical aspects of the study, I wanted to share five ideas/suggestions:

1. Meier et al. (2013) analyze recovered Nimbus I satellite imagery and highlight the possibility of a large Weddell Sea opening in 1964 on the basis of a polynya-like feature found in their imagery (see their Fig. 5). It would seem that your reconstructions (Fig. 6b) mostly exclude the possibility of a major, long-lasting polynya in that year. It would be interesting to discuss the relevance of your analysis to their findings.

2. Broecker et al. (1999) speculate that deep convection in Antarctic open-ocean polynyas must have supplied a greater amount of AABW during the Little Ice Age (~1350-1880) in order to meet present-day PO4* and 14C tracer budgets, with cessation of most open-ocean convection occurring during the post-Little Ice Age transition (possibly 1880-1945, or ongoing). It should be noted that aspects of this interpretation have since been challenged, e.g. by Orsi et al. (2001), who find bottom water ventilation rates from CFCs that seem to agree with those inferred from 14C distributions. There are also some intricacies involving differences in bottom water definitions. Nonetheless, is it possible that your reconstruction could shed some light on the conjecture raised by Broecker and colleagues, as well as their interesting suggestion that "ventilation of the deep Southern Ocean is episodic rather than steady"?

3. To the point in Lines 128-130 that "no high-resolution ocean sediment core that might provide a direct record of polynya activity is available": This unfortunately seems to be the case, as the most promising cores from ODP Sites 689/690 on Maud Rise have rather condensed Pleistocene sections due to low accumulation rates, and are possibly too dry at this point to yield useful samples. However, I anticipate that this study may well motivate future efforts to obtain new polynya proxy records, such as ocean sediment cores from sites with higher accumulation rates at Maud Rise or elsewhere. (I have been thinking about this, as have others!). I wonder if you could comment on how your method and resultant reconstruction of past polynya events from ice core records could complement or inform a similar effort using ocean sediment core records, which would likely cover millennial and longer scales, rather than the decadal-to-centennial

scales examined in this study.

4. I would be interested in seeing Lines 463-466 expanded to discuss alternative ice core chemical proxies in slightly more detail. Past work (e.g. Criscitiello et al. 2013) has attempted to reconstruct coastal polynya variability using sea salt aerosol proxies and is very relevant here. It might be useful to be specific as to which proxies might yield additional constraints on the open-ocean polynya reconstruction.

5. Resplandy et al. (2018) in Nature, cited on Line 79, has been retracted; the re-published version in Scientific Reports is Resplandy et al. (2019).

References:

Broecker, W. S., S. Sutherland, and T.-H. Peng, 1999: A possible 20th-century slowdown of Southern Ocean deep water formation. Science, 286, 1132–1135, doi:10.1126/science.286.5442.1132.

Campbell, E. C., E. A. Wilson, G. W. K. Moore, S. C. Riser, C. E. Brayton, M. R. Mazloff, and L. D. Talley, 2019: Antarctic offshore polynyas linked to Southern Hemisphere climate anomalies. Nature, 570, 319–325, doi:10.1038/s41586-019-1294-0.

Criscitiello, A. S., S. B. Das, M. J. Evans, K. E. Frey, H. Conway, I. Joughin, B. Medley, and E. J. Steig, 2013: Ice sheet record of recent sea-ice behavior and polynya variability in the Amundsen Sea, West Antarctica. J. Geophys. Res. Ocean., 118, 118–130, doi:10.1029/2012JC008077.

Meier, W. N., D. Gallaher, and G. G. Campbell, 2013: New estimates of Arctic and Antarctic sea ice extent during September 1964 from recovered Nimbus I satellite imagery. Cryosph., 7, 699–705, doi:10.5194/tc-7-699-2013.

Orsi, A. H., S. S. Jacobs, A. L. Gordon, and M. Visbeck, 2001: Cooling and ventilating the abyssal ocean. Geophys. Res. Lett., 28, 2923–2926, doi:10.1029/2001GL012830.

Resplandy, L., and Coauthors, 2018: Quantification of ocean heat uptake from changes

in atmospheric O2 and CO2 composition. Nature, 563, 105–108, doi:10.1038/s41586-018-0651-8.

– –, and Coauthors, 2019: Quantification of ocean heat uptake from changes in atmospheric O2 and CO2 composition. Sci. Rep., 9, 20244, doi:10.1038/s41598-019-56490-z.

---

## Referee Comment (RC1) · Anonymous Referee #1 · 14 Sep 2020

This paper is an interesting and relevant contribution to the discussion on whether large open-ocean polynyas in the Weddell Sea have occurred before the mid 1970s (the first and only such event in the instrumental (satellite-derived) record), and if yes, how regularly and with what intensity (size). According to the authors, sediment cores were so far useless for this purpose, while records from ice cores and Antarctic weather stations provide apparently more insight on this topic. With the further aid of atmosphere models that are being constrained at their surface by sea-ice concentration and thus sea-surface temperature, the authors make an attempt to connect phases of higher snow accumulation in ice cores "downwind of polynyas" with warm anomalies over the Weddell Sea, thereby projecting possible polynya occurrences over the past millen-

nium. The presented strategy and results are subject to substantial uncertainties. This has been clearly expressed throughout the paper. While the results thus need to be taken with caution, the methods and implications are nevertheless sound and worth publishing. Before doing so the authors may want to consider the following.

Main points:

⇒ In general, I think the text is too long for what is being presented. As an example, the Introduction, while providing a nice overview over the literature on Weddell Sea polynya formation, appears too long considering that the main thrust of this paper is polynya reconstruction from ice cores, and not the mechanism of polynya formation.

⇒ Section 2.3 is very technical and rather confusing (at least to me). I think a reader would get more out of it if the main steps of the procedure were displayed in a diagram.

⇒ At several occasions the authors mention (anomalous snow accumulation) "downwind of the polynya". Weijer et al. (2017; their Fig. 6) come up with an estimate of precipitation actually "downwind of a polynya" based on a high-resolution (0.10 degree sea and ocean; 0.25 degree atmosphere and land) CESM simulation (Small et al., 2014). While "just" a model result, if at all, land sees higher precipitation rates only when winds blow from northerly directions (NE or NW, N not shown). While your statement is thus supported by these simulation results, you make apparently no attempt to relate snow accumulation on land to wind direction. Is there any reason for why you do not take into account wind direction from ECHAM5-wiso or SPEAR in your reconstruction, or did I miss something?

⇒ The ice cores located around the Greenwich meridian at about 75S at altitudes higher than 2600 m (Fig.3a) do not seem to be impacted by any of the anomalies and regressions you are showing. There is also no physical explanation on how snow accumulation at such high altitudes and some 500 km inland could be affected by open-ocean polynyas. Including these ice cores in your reconstruction need a more convincing justification than just being in the (relative) geographic vicinity of potential

polynyas.

More detailed, line-by-line comments:

Line 23: Add "snow" before "accumulation.

Line 29: Coastal polynyas are additionally surrounded by land or ice shelves.

Lines 109 and 116: The two "Stössel et al." citations should be swapped.

Line 123: It seems more appropriate to replace "suggested" by "speculated". BTW: see also last paragraph of Kurtakoti et al. (2018) on this topic.

Line 127: "longer that" -> "that is longer than".

Lines 131-133: Awkward and too long a sentence. Polynyas may have an influence on the continent regardless of whether there is paleoclimate data for the specific ocean region available or not.

Line 157 and later: "associated to" -> "associated with".

Line 168: "dating error . . . maximum of a few years" doesn't sound very promising for reconstructing polynyas that last for only 2-3 years.

Lines 179-183: "A part of the trend could be due to a recent shift in polynya activity"; you could check that by reducing the time series to 1850-1980. How does the trend in general look like? Is there no trend before 1850? Why should removing the trend change the frequency of polynya occurrence?

Line 201: "They have constant forcing"; what forcing? Atmospheric CO2? Pre-industrial?

Line 202: "provide" -> "simulate"; insert "polynya" before "events".

Line 204: What does "model prior" mean? What does "their" refer to?

Line 209: The status of "Zhang et al., 2020" is submitted, so not accessible. So the

"differences between the two simulations" need to be described.

Fig.2: Why do you show annual-mean values rather than winter-mean or winter half-year values? In this region, polynyas do not exist in summer, and they exert a significant impact on the atmosphere only in winter. Wouldn't the explanation given in lines 269-270 be a good reason to just consider winter months?

Lines 277-278: "as for temperature . . . than the one in 1995"; what is this referring to? Fig.3b shows SMB, not temperature, and in Fig.2b, the temperature in the late 1970s is clearly the warmest of the shown record.

Line 301: Insert "-ocean" before "polynya".

Some suggested rewrite: Line 317: Insert "and defining" behind "calculating". Line 318: Insert "index" behind "S". Insert "-ocean" behind "Open". Line 321: "with this index" -> "onto the above specified mixed-layer depth index". Line 324: Insert "mixed-layer depth" before "index". Line 326: Insert "mixed-layer depth" before "index", and remove "based on the mixed layer depth".

Lines 333-334: "large warming and precipitation changes"; none occur at the high elevation ice cores along 75S around the Greenwich meridian shown in Fig.3a.

Lines 361-362: "a large fraction . . . higher than the mean"; Fig.3a shows 7 core sites, 4 of which with SMB values lower than the mean.

Line 390: "have preferred" -> "decided".

Lines 394-395: This sentence raises the concern that the uncertainties may make your conclusions obsolete.

Line 404: "show a clear maximum in 1975"; they also show a maximum in 1983 when there was no polynya.

Line 412: "downwind from the polynya"; why is this variable (wind) not considered in your reconstruction?

Line 447: Insert "atmosphere" before "model".

Line 450: "ice cores can be used" -> "it is tempting to use".

Line 453: "downwind": this has not been shown.

Line 458: "simple average" of what?; "data assimilation": what data has been assimilated?

Line 462: "of the index": what index?

Line 464: Add "Criscitiello et al., 2013"; see reference list in Ethan Campbell's comments.

Line 465: What does "these" refer to?

Lines 470-471: Or much larger polynyas, or indeed ice embayments in the Weddell Sea, as often simulated (see e.g. Cheon et al., 2014; Kurtakoti et al., 2018).

Line 473: What does "few" mean? 2-3 times?

---

## Author Comment (AC1) · 14 Sep 2020

We would like to thank Ethan Campbell for his nice words about our work and the interesting and useful comments. In order to follow more easily the discussion, his points are in italics below, our response in plain text.

*I want to thank the authors for submitting this fascinating work to Climate of the Past. The journal allows members of the broader scientific community to comment on preprints during the open discussion period. I thought I'd use the opportunity to provide a few brief thoughts, which are not intended to be an exhaustive review. As someone with a close interest in the Weddell polynya phenomenon (e.g. Campbell et al. 2019), I found this study to be particularly exciting. The techniques used by the authors to identify the fingerprints of past polynyas in ice core records of surface mass balance (among other data) seem well-motivated and promising. Without commenting further on the technical aspects of the study, I wanted to share five ideas/suggestions:*

*1. Meier et al. (2013) analyze recovered Nimbus I satellite imagery and highlight the possibility of a large Weddell Sea opening in 1964 on the basis of a polynya-like feature found in their imagery (see their Fig. 5). It would seem that your reconstructions (Fig.6b) mostly exclude the possibility of a major, long-lasting polynya in that year. It would be interesting to discuss the relevance of your analysis to their findings.*

We are sorry that we missed the reference to the study of Meier et al. (2013) and their results will be discussed in the revised version of the manuscript. Meier et al. (2013) present indications of the presence of reduced ice concentration in the region of the formation of the Weddell polynya in September 1964 but, because of the small number of available observations, they state that 'However, it is not clear if there was a polynya at or near the time of the image or just an indication of leads and clouds.' Our methodology is designed to identify large open ocean polynyas, as in 1974-1976. Short events have a too small imprint to distinguish them in ice core records. We cannot conclude from our results if a polynya was present or not in September 1964, but our reconstruction indeed suggests that no event similar to the opening in 1974-1976 occurred at that time.

*2. Broecker et al. (1999) speculate that deep convection in Antarctic open-ocean polynyas must have supplied a greater amount of AABW during the Little Ice Age(~1350-1880) in order to meet present-day PO4* and 14C tracer budgets, with cessation of most open-ocean convection occurring during the post-Little Ice Age transition(possibly 1880-1945, or ongoing). It should be noted that aspects of this interpretation have since been challenged, e.g. by Orsi et al. (2001), who find bottom water ventilation rates from CFCs that seem to agree with those inferred from 14C distributions. There are also some intricacies involving differences in bottom water definitions. Nonetheless, is it possible that your reconstruction could shed some light on the conjecture raised by Broecker and colleagues, as well as their interesting suggestion that"ventilation of the deep Southern Ocean is episodic rather than steady"?*

The potential impact of centennial-scale changes in open ocean polynya formation on deep water formation is a strong motivation for our work, as discussed in the introduction. Nevertheless, our results do not address specifically this point at this stage. We did not want to speculate too much on this subject in the submitted

manuscript as we had not much specific information to add and thus prefer to leave this for subsequent studies. On a more general point, the magnitude of the changes in ocean circulation over the past millennium as well as their impact are very uncertain. For instance, in a previous study (Goosse 2018), the transport of anomalies by the mean circulation in the Southern Ocean has been shown to explain the timing of some warm events before the LIA (based on mechanisms similar to the ones proposed for the more recent past, see Marshall et al., 2015 or Armour et al. 2016) but I was not able to identify a contribution of changes in the circulation.

*3. To the point in Lines 128-130 that "no high-resolution ocean sediment core that might provide a direct record of polynya activity is available": This unfortunately seems to be the case, as the most promising cores from ODP Sites 689/690 on Maud Rise have rather condensed Pleistocene sections due to low accumulation rates, and are possibly too dry at this point to yield useful samples. However, I anticipate that this study may well motivate future efforts to obtain new polynya proxy records, such as ocean sediment cores from sites with higher accumulation rates at Maud Rise or elsewhere. (I have been thinking about this, as have others!). I wonder if you could comment on how your method and resultant reconstruction of past polynya events from ice core records could complement or inform a similar effort using ocean sediment core records, which would likely cover millennial and longer scales, rather than the decadal-to-centennial scales examined in this study.*

That would be great if our study could motivate the collection of new records allowing to refine the reconstructions of past polynya activity. We can anticipate significant practical problems, related to the recovery of a high-resolution oceanic core in the region, its interpretation, the maybe small overlap with the ice cores sensitive to open ocean polynya formation (some of them cover only a few centuries), etc. However, there is no fundamental problem to combine different records using the method we have applied. This is a very interesting perspective, in particular as the ocean sediment core records could constrain the low frequency variations that may be difficult to identify from the ice core records and we will mention it in the revised version of our manuscript.

*4. I would be interested in seeing Lines 463-466 expanded to discuss alternative ice core chemical proxies in slightly more detail. Past work (e.g. Criscitiello et al. 2013) has attempted to reconstruct coastal polynya variability using sea salt aerosol proxies and is very relevant here. It might be useful to be specific as to which proxies might yield additional constraints on the open-ocean polynya reconstruction.*

The intention in the submitted manuscript was not to go in the details on this point. This is the reason why we just cite review studies focusing on the reconstruction of the sea ice (or of its absence in the case of a polynya). Without a deeper investigation, it is hard to guess which proxy will provide the largest constraint but sodium and chlorine content in ice cores (sea salt) are very good candidates. The source of sea salt aerosol to the polar atmosphere strongly depends on the condition at ocean surface, specifically the presence of sea ice. The formation of an open ocean polynya in winter should thus have a large impact on the sea salt transferred to the atmosphere in the Weddell Sea. This signal can then be transported to the continent by the winds and recorded in ice cores. Although biological activity in the polynya itself may be limited in

winter, the formation of the open ocean polynya can have an impact of the vertical structure of the water column, on light availability, on the nutrient input at surface and thus on biological production later in the season (e.g., von Berg et al. 2020). This might also be recorded in ice cores, for instance in their Methanesulfonic acid content. Those proxies have been used to reconstruct coastal polynyas (e.g., Rhodes et al. 2009, Criscitiello et al. 2013; Mezgec et al. 2018). We are not aware of any study analyzing the link between open ocean polynya activity and those chemical records, so this deserves more investigation. However, as suggested, this point will be developed in the revised version of the manuscript.

*5. Resplandy et al. (2018) in Nature, cited on Line 79, has been retracted; the re-published version in Scientific Reports is Resplandy et al. (2019).*

Thanks a lot for pointing this. We will cite the updated version in the revised version of the manuscript.

References:

Armour, K.C., Marshall, J., Scott; J.R., Donohoe, A., Newsom, E.R., Southern Ocean warming delayed by circumpolar upwelling and equatorward transport. Nat. Goes., DOI: 10.1038/NGEO2731, 2016.

Broecker, W. S., Sutherland, S., and Peng, T.-H.: A possible 20th-century slowdown of Southern Ocean deep water formation. Science, 286, 1132–1135,doi:10.1126/science.286.5442.1132, 1999.

Campbell, E. C., Wilson, E. A., Moore, G. W. K., Riser, S. C., Brayton, C. E., Mazloff, M. R.,and Talley, L. D.: Antarctic offshore polynyas linked to Southern Hemisphere climate anomalies. Nature, 570, 319–325, doi:10.1038/s41586-019-1294-0, 2019.

Criscitiello, A. S., Das, S. B., Evans, M. J., Frey, K. E., Conway, H., Joughin, I., Medley, B., and Steig, E. J.: Ice sheet record of recent sea-ice behavior and polynya variability in the Amundsen Sea, West Antarctica. J. Geophys. Res. Ocean, 118, 118–130,doi:10.1029/2012JC008077, 2013.

Goosse, H., 2017: Reconstructed and simulated temperature asymmetry between continents in both hemispheres over the last centuries. Clim. Dyn. 48, 1483–1501, doi:10.1007/s00382-016-3154-z.

Levine, J. G., Yang, X., Jones, A. E., Wolff, E. W.: Sea salt as an ice core proxy for past sea ice extent: A process based model study, J. Geophys. Res. Atmos., 119, 5737–5756, doi:10.1002/2013JD020925, 2014.

Marshall, J., Scott, J.R., Armour, K.C., Campin, J.-M., Kelley, M., Romanou, A.: The ocean's role in the transient response of climate to abrupt greenhouse gas forcing. Clim. Dyn. 4, 2287-2299, 2015

Meier, W. N., Gallaher, D., and Campbell, G. G.: New estimates of Arctic and Antarctic sea ice extent during September 1964 from recovered Nimbus I satellite imagery. Cryosph., 7, 699–705, doi:10.5194/tc-7-699-2013, 2013.

Mezgec, K., Stenni, B., Crosta, X., Masson-Delmotte, V., Baroni, C., Braida, M., Ciardini, V., Colizza, E., Melis, R., Salvatore, M.C., Severi M., Scarchilli C., Traversi R., Udisti R. and Frezzotti M.: Holocene sea ice variability driven by wind and polynya efficiency in the Ross Sea. Nat. Commun., 8, 1334, 2017.

Orsi, A. H., Jacobs, S. S., Gordon, A. L., and Visbeck, M.: Cooling and ventilating the abyssal ocean. Geophys. Res. Lett., 28, 2923–2926, doi:10.1029/2001GL012830, 2001.

Rhodes, R.H., Bertler, N.A.N., Baker, J.A., Sneed, S.B., Oerter, H., Arrigo, K.R.; Sea ice variability and primary productivity in the Ross Sea, Antarctica, from methylsulphonate snow record. Geophys. Res. Lett., 36, L10704. doi:10.1029/2009GL037311, 2009

Rhodes, R. H., Yang, X., Wolff, E.W.: Sea ice versus storms: What controls sea salt in Arctic ice cores? Geophysical Research Letters, 45, 5572–5580. https://doi.org/10.1029/2018GL077403, 2018.

Resplandy, L., and Coauthors : Quantification of ocean heat uptake from changes in atmospheric $O_2$ and $CO_2$ composition. Nature, 563, 105–108, doi:10.1038/s41586-018-0651-8, 2018.

– –, and Coauthors : Quantification of ocean heat uptake from changes in atmospheric $O_2$ and $CO_2$ composition. Sci. Rep., 9, 20244, doi:10.1038/s41598-019-56490-z, 2019.

von Berg, L., Prend, C. J., Campbell, E. C., Mazloff, M. R., Talley, L. D., and Gille, S. T.: Weddell Sea phytoplankton blooms modulated by sea ice variability and polynya formation. Geophysical Research Letters, 47, e2020GL087954. https://doi.org/10.1029/2020GL087954, 2020

---

## Referee Comment (RC2) · Anonymous Referee #2 · 21 Sep 2020

General Comments

This article is an interesting and valuable contribution to our ability to determine the frequency of large, multi-year open-ocean polynyas in the Weddell Sea. The authors use a combination of continental observations and atmospheric model simulations to identify the potential signature from these open-ocean polynyas in continental ice cores located between 50 oW and 0 oE. The authors then use a series of high-resolution ice core records to estimate when large Weddell Sea open-ocean polynyas have occurred during the 800 yrs prior to the satellite era. There are substantial uncertainties associated with the reconstruction methods utilised here, which are clearly stated and

discussed throughout the paper.

The paper presents novel techniques for identifying past open-ocean polynyas which are worth publishing. However, the uncertainties result in the majority of the conclusions being largely speculative. Therefore, this paper would benefit from a refocused discussion on the possible wider implications of the predicted polynya occurrence frequency, as these implications would help guide future work into corroborating or disproving the polynya frequency predicted by the authors.

Specific Comments

=> The introduction is very long, with substantial detail on polynya formation that seems superfluous to the focus and aims of the paper. If more discussion were to be added on the link between the formation mechanisms and the predicted occurrences in Figure 6 then this detail would become more relevant to the scope of the paper.

=> In both Figures 3a and 4 there are anomalies across large parts of the West Antarctic Peninsula and Amundsen-Bellingshausen Seas regions. The authors indicate in line 274 that these anomalies are not related to the Weddell Sea polynya. However, there is no further discussion of what is causing these anomalies. It is important to at least speculate as to what is causing these anomalies and, crucially, whether it is also responsible for the anomalies closer to the polynya that are currently being interpreted as a signal from the polynya itself.

=> Concerning the Hadley Centre data set used to drive the ECHAM5-wiso atmospheric model, it is not clear which years have been used in this study. Lines 189-190 suggest that data from 1871-2011 has been used whereas Figure 4 only seems to have the years 1958-2000. For either option there needs to be an appreciation of the limitations in using the Antarctic sea ice data from Rayner et al. (2003) to run the model. For example, the Antarctic sea-ice extents for the years 1871-1927 are all duplications of the 1927 climatology.

=> There should be some discussion of how changes in wind direction could influence the variability seen in Figures 6 and 7 as well as possible explanations for the hypothesised centennial-scale variability in line 432.

Technical Corrections

Lines 55-57: statements on the ocean preconditioning should be referenced.

Line 229: replace "averages is" with "averages are".

Line 357: presumably should be 50 oW not 50 oE, as in line 368.

Line 399: should be Figure 5 not Figure 4.

Lines 425-428: it is unclear whether this is referring to years when any single index is greater than 0.8/1 or when the average of all the indices is greater than 0.8/1.

Figure 6: the use of "complete" and "all" to identify different time ranges is confusing for the reader.

Figure 6: all the overlapping coloured records make it difficult to identify years with high values in all the indices, especially in Figure 6a. The addition of arrows to indicate which years had an average index value greater than 0.8 or 1 would be beneficial for analysing the changes in polynya frequency during the last millennium.

---

## Author Comment (AC3) · 16 Oct 2020

We would like to thank the reviewer for the careful evaluation of our work and for the constructive comments. In the following, the reviewer's points are in italics, our response in plain text.

General Comments

*This article is an interesting and valuable contribution to our ability to determine the frequency of large, multi-year open-ocean polynyas in the Weddell Sea. The authors use a combination of continental observations and atmospheric model simulations to identify the potential signature from these open-ocean polynyas in continental ice cores located between 50°W and 0°E. The authors then use a series of high-resolution ice core records to estimate when large Weddell Sea open-ocean polynyas have occurred during the 800 yrs prior to the satellite era. There are substantial uncertainties associated with the reconstruction methods utilised here, which are clearly stated and discussed throughout the paper.*

*The paper presents novel techniques for identifying past open-ocean polynyas which are worth publishing. However, the uncertainties result in the majority of the conclusions being largely speculative. Therefore, this paper would benefit from a refocused discussion on the possible wider implications of the predicted polynya occurrence frequency, as these implications would help guide future work into corroborating or disproving the polynya frequency predicted by the authors.*

We agree with the reviewer that the uncertainties are large. This will be expanded in the revised version, in particular by presenting the reconstructions differently (see below) and by discussing more explicitly the perspective to reduce the uncertainties, as suggested, in new paragraphs of the final section (Discussion and Conclusions)

Specific Comments

*=> The introduction is very long, with substantial detail on polynya formation that seems superfluous to the focus and aims of the paper. If more discussion were to be added on the link between the formation mechanisms and the predicted occurrences in Figure 6 then this detail would become more relevant to the scope of the paper.*

As suggested by the reviewer, we will add in the final section (Discussion and Conclusions) new paragraphs to discuss the link between the mechanisms of formation and the predicted occurrence of open ocean polynya, making the connection to the material described in the introduction. Specifically, this final section will include a discussion of the wind changes, in particular how a connection between our reconstructions of polynya activity can be made with available reconstructions of the Southern Annular Mode (SAM; related to the intensity and the position of the Westerlies) for the past millennium. This will justify the discussion of the hypotheses related to the role of SAM in polynya formation in the introduction. The link to oceanic processes mentioned in the introduction is a bit more difficult to develop in the final section because of the small number of high resolution data available, but we will add an explanation of the potential impact of the reconstructed changes on the ocean state and how new information about past ocean changes could support, or refute some of the changes in polynya occurrence estimated from our reconstruction. Additionally, we propose to shorten the introduction by about 20%, to focus on the elements specifically relevant to the other sections of the manuscript.

*=> In both Figures 3a and 4 there are anomalies across large parts of the West Antarctic Peninsula and Amundsen-Bellingshausen Seas regions. The authors indicate in line 274 that these anomalies are not related to the Weddell Sea polynya. However, there is no further discussion of what is causing these anomalies. It is important to at least speculate as to what is causing these anomalies and, crucially, whether it is also responsible for the anomalies closer to the polynya that are currently being interpreted as a signal from the polynya itself.*

We agree that the sentence line 274 was a bit too short and that the topic deserves a more detailed explanation. We propose to change this line from 'it is unlikely that they are all related to the great Weddell Sea polynya formation' to ' it is unclear if they are all related to the great Weddell Sea polynya formation' and discuss further this point in section 4. We propose to add, in the section devoted to the description of the fingerprint of the polynya, a justification as to why we consider that the higher accumulation in the section roughly between 50°W and 0° is a more robust signature of the open ocean polynya formation compared to other areas of the ice sheet.

*=> Concerning the Hadley Centre data set used to drive the ECHAM5-wiso atmospheric model, it is not clear which years have been used in this study. Lines 189-190 suggest that data from 1871-2011 has been used whereas Figure 4 only seems to have the years 1958-2000. For either option there needs to be an appreciation of the limitations in using the Antarctic sea ice data from Rayner et al. (2003) to run the model. For example, the Antarctic sea-ice extents for the years 1871-1927 are all duplications of the 1927 climatology.*

The Rayner et al. (2003 and updates) estimate of the Antarctic sea ice concentration has indeed clear limitations, with could potentially lead to large uncertainties in the ECHAM5-wiso results. The simulation covers the period 1871-2011, as mentioned in section 2.2. However, we only use the period 1958-2000 here to obtain a climatology of the model results that we compare with the years 1974-1976 corresponding to the great Weddell polynya existence. For clarity, this will be specified in the revised version when we describe the simulation itself. We do not use the earlier period, in particular the years 1871-1927 mentioned by the referee when the uncertainties on the sea ice extent are particularly large. The uncertainties are already large before 1973 as no satellite-based imagery is directly included and the reconstruction is derived from various climatologies. We have chosen the years 1958-2000 as a reference to have a common period for all the datasets in Figures 2, 3 and 4. Nevertheless, our results for the region of the ice sheet close to the Weddell Sea, and therefore more directly influenced by the polynya formation, are not very sensitive to the reference period chosen. This is illustrated in the figure below (Figure R2.1) where the reference period is 1979-2011 (when satellite information is available and the uncertainties on the Rainer et al. (2003) data set are lowest).

[Figure]

[Figure]

Figure R2.1: Anomaly of (a) annual mean temperature (°C), (b) precipitation (mm w.e./y), averaged over 1974-1976 compared to the period 1979-2011 in a simulation performed with ECHAM5-wiso. This figure is similar to Figure 4 of the main manuscript but with a different reference period.

*=> There should be some discussion of how changes in wind direction could influence the variability seen in Figures 6 and 7 as well as possible explanations for the hypothesised centennial-scale variability in line 432.*

As suggested, we will include a discussion of the possible role of wind changes, in particular related to the Southern Annular Mode, in the variability seen in our reconstruction. See also the response to the main comment above.

Technical Corrections

*Lines 55-57: statements on the ocean preconditioning should be referenced.*

References will be added.

*Line 229: replace "averages is" with "averages are".*

Corrected.

*Line 357: presumably should be 50°W not 50°E, as in line 368.*

Thank you for pointing out this mistake. It will be corrected in the revised version.

*Line 399: should be Figure 5 not Figure 4.*

This will be corrected.

*Lines 425-428: it is unclear whether this is referring to years when any single index is greater than 0.8/1 or when the average of all the indices is greater than 0.8/1.*

The discussion is written for any individual reconstruction. This will be specified in the revised version.

*Figure 6: the use of "complete" and "all" to identify different time ranges is confusing for the reader.*

Instead of the reconstructions based on the 6 ice core records ('all') or only on the records that cover the entire studied period (i.e. 1250-now); 'complete'), we propose to estimate the uncertainties of the reconstructions by showing the reconstructions obtained when selecting five out of the 6 records in turn for the assimilation. This is a more comprehensive and objective way to measure the uncertainties (e.g. Hakim et al., 2016). The word 'complete' and 'all" will thus not be used anymore in the revised version.

*Figure 6: all the overlapping coloured records make it difficult to identify years with high values in all the indices, especially in Figure 6a. The addition of arrows to indicate which years had an average index value greater than 0.8 or 1 would be beneficial for analysing the changes in polynya frequency during the last millennium.*

We propose to modify the way the reconstructions are presented and their uncertainty estimated by showing for the three methods (simple average and data assimilation using SPEAR_AM2 and SPEAR_LO) only one curve, that uses all six ice core records. For those three reconstructions, the uncertainties will be estimated by the range in the results from assimilating only 5 out of the 6 ice core records using the same methodology (a different record is excluded each time). This will have the advantage of reducing the number of curves per panel and thus providing clearer figures. We will also add horizontal lines for values of 0.8 and 1.0 to clearly see when each threshold is crossed (the other horizontal bars will be removed).

References

Hakim, G. J., J. Emile-Geay, E. J. Steig, D. Noone, D. M. Anderson, R. Tardif, N. Steiger, and W. A. Perkins (2016), The last millennium climate reanalysis project: Framework and first results, J. Geophys. Res. Atmos., 121, 6745–6764, doi:10.1002/2016JD024751

---

## Author Response (AR1)

We would like to thank the Editor for the careful and constructive evaluation of our work. Our responses to the comments are included below.

*The manuscript has received two reviews and a public comment. All three have commented on the importance of the work and made constructive comments on ways to improve the manuscript. I thank the authors for responding to each of the comments in turn, where they outline how these suggestions could be incorporated into a revised manuscript. I note that the reviewers are very clear that the text gives a clear discussion of the substantial uncertainties associated with the approach which is being tested here. The responses to the reviewers comments indicate that the authors will address those areas where a lack of clarity was present in the submitted manuscript, including some revisions to the graphics.*

We have implemented in the revised version the modifications described in the responses to the reviewers that were submitted online at the end of the discussion phase. In particular, we have clarified some points, insisted on some of the limitations of the study and revised figures 5 and 6 to improve their readability and show more clearly the uncertainties.

*Both reviewers commented on the length of the Introduction. In the replies, the authors make a strong case for why this information is still required. I agree with both parties. The authors indicate that they will reduce the Introduction by ~20%, which may or may not address the reviewers concerns given the current length of this section (~ 4 pages currently). I suggest that the authors may also wish to consider splitting the text into e.g. a short introduction which lays out the rationale for this study and its approach, followed by a section on polynya formation and impacts? Alternatively, as in the replies to Reviewer 2, consider whether some of this detail may be better placed when evaluating the findings in the Discussion (see response to Reviewer 2 point 1).*

This is a very good suggestion. We have moved the discussion of the role of the winds and the impact of polynya formation on the ocean state initially in the introduction to the last section where the potential link between our reconstruction and other records is evaluated (lines 477-504). This reduces further the introduction compared to what we initially proposed. The introduction is now a bit more than 2-page long, a reduction of nearly 40% compared to the previously submitted version.

Response to Reviewer 1

We would like to thank the reviewer for the careful evaluation of our work and for the constructive comments. In the following, the reviewer's points are in italics, our response in plain text. When it is not specified otherwise, the line numbers refer to the revised version without track changes.

*This paper is an interesting and relevant contribution to the discussion on whether large open-ocean polynyas in the Weddell Sea have occurred before the mid 1970s (the first and only such event in the instrumental (satellite-derived) record), and if yes, how regularly and with what intensity (size). According to the authors, sediment cores were so far useless for this purpose, while records from ice cores and Antarctic weather stations provide apparently more insight on this topic. With the further aid of atmosphere models that are being constrained at their surface by sea-ice concentration and thus sea-surface temperature, the authors make an attempt to connect phases of higher snow accumulation in ice cores "downwind of polynyas" with warm anomalies over the Weddell Sea, thereby projecting possible polynya occurrences over the past millennium. The presented strategy and results are subject to substantial uncertainties. This has been clearly expressed throughout the paper. While the results thus need to be taken with caution, the methods and implications are nevertheless sound and worth publishing. Before doing so the authors may want to consider the following.*

Main points:

⇒*In general, I think the text is too long for what is being presented. As an example, the Introduction, while providing a nice overview over the literature on Weddell Sea polynya formation, appears too long considering that the main thrust of this paper is polynya reconstruction from ice cores, and not the mechanism of polynya formation.*

We agree that the introduction is long but one of the goals of the paper is to provide motivation for the reconstruction of past polynya activity. Consequently, we consider that it is important to review current knowledge on the mechanisms of polynya formation and on the impact of polynya opening. In our opinion, discussing existing modeling results is also needed as, on the one hand, a goal of the reconstruction is the validation of the frequency of polynya formation in models and, on the other hand, the spatial pattern provided by some models are an important source of information for the reconstruction. Describing the mechanisms behind polynya opening is also interesting as they could be linked to the

fingerprint of the polynya in the system and thus useful to compare our reconstructions with independent observations. This aspect is developed in the revised version, in particular by adding two paragraphs in the final section on this issue (lines 477-504). We have also moved some parts of the introduction to this

55  final paragraph as suggested by the Editor. Finally, we have removed the paragraphs that were not strongly connected with the material discussed later in the paper, resulting in a total reduction of the length of the introduction by nearly 40%.

⇒*Section 2.3 is very technical and rather confusing (at least to me). I think a reader would get more out of it if the main steps of the procedure were displayed in a diagram.*

60  The data assimilation technique is strictly identical to the one applied in several previous studies in which extensive descriptions are available. This is the reason why we only gave here a short overview of the methodology itself, the majority of the section being devoted to the way the data and their errors are handled as this is specific to the present work.

Nevertheless, we understand that it may be difficult to follow the goals and interest of the method from

65  the short paragraph, lines 221-227 of the submitted version, for readers who are not familiar with data assimilation and thus simply citing previous work may be not sufficient.

We have thus expanded significantly the description of the methodology in the revised version to provide all the needed information to understand the results presented later on in the manuscript (lines 190-203). We prefer this solution to adding a diagram as suggested, as figures describing data assimilation must

70  remain very general to be easily understood, and thus cannot include all the specificities that could be included in a text.

⇒*At several occasions the authors mention (anomalous snow accumulation) "down-wind of the polynya". Weijer et al. (2017; their Fig. 6) come up with an estimate of precipitation actually "downwind of a polynya" based on a high-resolution (0.10 degree sea and ocean; 0.25 degree atmosphere and land)*

75  *CESM simulation (Small et al., 2014). While "just" a model result, if at all, land sees higher precipitation rates only when winds blow from northerly directions (NE or NW, N not shown). While your statement is thus supported by these simulation results, you make apparently no attempt to relate snow accumulation on land to wind direction. Is there any reason for why you do not take into account wind direction from ECHAM5-wiso or SPEAR in your reconstruction, or did I miss something?*

80 The reconstruction is based on annual time series, because of the resolution of the ice cores. At the daily-scale, we could link the snow accumulation on land with the wind direction in the models as in Weijer et al. (2017). However, the signal that we compare with the records is the total snow accumulation over one year. This is the reason why we show those maps on figure 4 and 5. As shown in Fig. R1.1, the winds over the polynya area and near the coast are on annual average directed mainly westward. This is the

85 reason why we suggested that the signal from the polynya on the continent should be seen more to the west of the polynya than to the east of the polynya. Nevertheless, it is not possible to analyze the relationship between wind and precipitation at the annual scale as the majority of precipitation on land could be mainly due to winds coming from the north (NE or NW), as indicated by the Reviewer, that lasted only for a few hours or days. This is specified more clearly in the revised version of the manuscript,

90 insisting more explicitly on the interest of the work of Weijer et al. (2017) for our interpretation when we discuss the model results in section 3 (lines 270-275) and section 4 (lines 344-346). We also insist that we must focus on model results at the annual scale because of the annual resolution of the ice cores (lines 228-229).

95

[Figure]

Figure R1.1. Annual mean winds in the Atlantic sector of the Southern Ocean in the ERA-5 reanalysis, averaged over the period 1990-2019 (Hersbach et al. 2020).

⇒*The ice cores located around the Greenwich meridian at about 75S at altitudes higher than 2600 m (Fig.3a) do not seem to be impacted by any of the anomalies and regressions you are showing. There is also no physical explanation on how snow accumulation at such high altitudes and some 500 km inland could be affected by open-ocean polynyas. Including these ice cores in your reconstruction need a more convincing justification than just being in the (relative) geographic vicinity of potential polynyas.*

In order to see more clearly the location of the ice cores compared to the changes observed and simulated for periods of polynya opening, the location of the cores has been added on Figs. 4b and Fig. 5cd. This shows clearly that none of the ice cores are located in the region where the largest changes (in magnitude) are simulated. However, for a majority of the ice cores, they are located in a region where a positive change is reconstructed (on Figure 3, the 6 selected cores are in the regions where positive changes are observed) or simulated (positive values are obtained at the 6 locations for the two SPEAR model versions, and at 3 locations for the ECHAM5-wiso simulations). The signal of the polynya is stronger at the coast than inland, but the mean accumulation and variability is also higher there compared to more inland locations, potentially introducing more noise (one of the negative values of ECHAM5-wiso is a coastal site while the ice core at the same location has a clear and strong positive signal for the period of the polynya, i.e. for the same years). Our goal here is not to focus on the physical mechanisms that may explain the signal inland at relatively high elevation associated with polynyas but, for instance, it has been shown that storms coming from the Southern Ocean can propagate far inland and be responsible for high accumulation events (e.g., Turner et al., 2019), with potentially an influence of the polynya region on those storms (see also Wang et al.,2020, for a general evaluation of the impact of sea ice changes on snow accumulation over the Antarctic ice sheet). In addition to the modifications of Figures 4 and 5, we have expanded our section devoted to the selection of the ice cores, explaining our choices more clearly (lines 347-363). Furthermore, the impact of the choice of the ice core records on our results is investigated in the revised version by comparing the reconstruction based on all six records with six reconstructions obtained by selecting only five of the six records. As it is difficult to assess quantitatively which of the ice cores are the best ones, we consider that it is the most objective choice to retain a maximum of ice core records while estimating the uncertainty due to the ice core selection.

*More detailed, line-by-line comments:*

*Line 23: Add "snow" before "accumulation.*
130 Modified as suggested.

*Line 29: Coastal polynyas are additionally surrounded by land or ice shelves.*
This sentence was not general enough. We propose to modify by 'Polynyas are ice-free oceanic areas within the sea-ice pack'.

135

*Lines 109 and 116: The two "Stössel et al." citations should be swapped.*
The citation has been swapped as suggested.

*Line 123: It seems more appropriate to replace "suggested" by "speculated". BTW:see also last*
140 *paragraph of Kurtakoti et al. (2018) on this topic.*
We have replaced the word "suggested" as indicated and add the reference to Kurtakoti et al. (2018) but without more discussion to avoid making the introduction even longer.

*Line 127: "longer that" -> "that is longer than".*
145 Modified as suggested.

*Lines 131-133: Awkward and too long a sentence. Polynyas may have an influence on the continent regardless of whether there is paleoclimate data for the specific ocean region available or not.*
We propose to remove the first half of the sentence, which repeats and summarizes the message from the
150 previous paragraph.

*Line 157 and later: "associated to" -> "associated with".*
This has been corrected in this line and each time 'associated' is used.

155 *Line 168: "dating error...maximum of a few years" doesn't sound very promising for reconstructing polynyas that last for only 2-3 years.*
We agree with the reviewer. The dating uncertainty puts a strong constraint on our results. It is one of the reasons why we smoothed the records before our analyses. This issue was mentioned in the submitted version and we insist even more on this in the conclusion of the revised version to make sure that the
160 reconstruction limitations are not underestimated (e.g., lines 425-428; 454-456, line 515).

*Lines 179-183: "A part of the trend could be due to a recent shift in polynya activity"; you could check that by reducing the time series to 1850-1980. How does the trend in general look like? Is there no trend before 1850? Why should removing the trend change the frequency of polynya occurrence?*
165 There is no clear trend in the records before 1850 but some of them display increasing precipitation over the past century (e.g. Medley et al., 2018). The origin of this trend is debated but looking at the records themselves, at first sight, it appears as more of a sustained precipitation increase rather than a trend due to some changes in the occurrence of events such as the ones associated with polynya formation. To avoid

misinterpretation, and as this trend does not seem to be easily related to polynya activity, we prefer to remove it and acknowledge the limitation this imposes on the interpretation of possible changes in the frequency of polynya formation. If this trend is not removed, the number of polynya events decreases for the second half of the 19[th] century as mean accumulation was lower back then and thus the likelihood to overstep the threshold corresponding to polynya formation in our reconstruction becomes weaker (Figure R1.2).

[Figure]

Figure R1.2. Index of polynya activity based on 5 surface mass balance records using a simple average of standardized time series with detrending as in figure 6 of the submitted manuscript (black curve) and without detrending (red curve). The times series have all been scaled to have a value of 1 in 1975

*Line 201: "They have constant forcing"; what forcing? Atmospheric CO2? Pre-industrial?*
Yes, this is pre-industrial conditions. This is specified in the revised manuscript.

*Line 202: "provide" -> "simulate"; insert "polynya" before "events".*
Modified as suggested.

*Line 204: What does "model prior" mean? What does "their" refer to?*
The prior is the distribution of the initial estimates of the system, thus the model ensemble, before using the observations to constrain the results. At each time-step of the data assimilation, the information

provided by the prior is updated according to the available observations. This term is now defined in the method section in the revised version (see our response to the main comments).

*Line 209: The status of "Zhang et al., 2020" is submitted, so not accessible. So the differences between*
195 *the two simulations" need to be described.*
As the status of Zhang et al. (2020) has not changed since submission, we do not cite this paper but rather two papers that describe the simulations (SPEAR_LO: Delworth et al. (2020) for SPEAR_LO and Zhang et al. ( 2019) for SPEAR_AM2).

200 *Fig.2: Why do you show annual-mean values rather than winter-mean or winter half-year values? In this region, polynyas do not exist in summer, and they exert a significant impact on the atmosphere only in winter. Wouldn't the explanation given in lines 269-270 be a good reason to just consider winter months?*
We show the annual mean because the ice cores provide estimates of annual mean precipitation. In order to detect the effect of the polynya from ice core records, its opening should thus have a fingerprint on
205 annual mean variables that is then extracted using the methods proposed here. This is why we show annual mean results. The signal in the polynya region is stronger in winter (for instance compare Figure R1.3 with Figure 4 of the manuscript) but it is large and clear enough during this season that it can be seen on the annual mean too. Although the amplitude is smaller for the annual mean compared to the winter mean, the two patterns are clearly similar in the region of interest.

[Figure]

210

Figure R1.3. Anomaly of (a) winter (JJA) mean temperature (°C), (b) precipitation (mm w.e./y), (c) mean $\delta^{18}O$ of precipitation (‰) and (d) mean $\delta^{18}O$ weighted by the precipitation amount averaged over 1974-1976 compared to the period 1958-2000 in a simulation performed with ECHAM5-wiso (‰). This figure is the equivalent of Figure 4 of the main
215    manuscript that displays the annual mean anomaly.

*Lines 277-278: "as for temperature...than the one in 1995"; what is this referring to?Fig.3b shows SMB, not temperature, and in Fig.2b, the temperature in the late 1970s is clearly the warmest of the shown record.*

220 Sorry for the confusion, we are referring to SMB (Fig 3b), but wanted to compare to the temperature signal. As the temperature is already discussed in the previous paragraph, we propose to remove the words 'as for temperature' to make the sentence simpler and focused on SMB.

*Line 301: Insert "-ocean" before "polynya".*

225 Modified as suggested.

*Some suggested rewrite: Line 317: Insert "and defining" behind "calculating". Line 318: Insert "index" behind "S". Insert "-ocean" behind "Open". Line 321: "with this index" -> "onto the above specified mixed-layer depth index". Line 324: Insert "mixed-layer depth" before "index".*

230 Modified as suggested, except for the comment "*Line 318: Insert "index" behind "S".*" as the word index is already once in the sentence and the meaning appeared clear to us.

*Line 326: Insert "mixed-layer depth" before "index", and remove "based on the mixed layer depth".*
Modified as suggested.

235

*Lines 333-334: "large warming and precipitation changes"; none occur at the high elevation ice cores along 75S around the Greenwich meridian shown in Fig.3a.*
See main comment.

240 *Lines 361-362: "a large fraction...higher than the mean"; Fig.3a shows 7 core sites, 4 of which with SMB values lower than the mean.*
We are sorry, one of the ice cores was not visible in the figure, as it was hidden by another core in close spatial proximity. Additionally, the sentence was not referring to all the ice cores shown in Fig. 3a but only the ones selected in the reconstruction (from which 4 out of 6 have positive values). This has been
245 changed in the revised version.

*Line 390: "have preferred" -> "decided".*
Modified as suggested.

250 *Lines 394-395: This sentence raises the concern that the uncertainties may make your conclusions obsolete.*
We agree that our reconstructions have strong limitations, which we mentioned in the submitted manuscript. We have made this point even stronger in the revised manuscript to state clearly where are the uncertainties and to underline the few robust conclusions that we could nevertheless gain from our
255 analyses as well as the perspectives to further reduce uncertainties in follow-up studies.

*Line 404: "show a clear maximum in 1975"; they also show a maximum in 1983 when there was no polynya.*

The maximum in 1983 is discussed 3 lines below in the text. We propose to remove 'clear' to not overemphasize the maximum in 1975.

*Line 412: "downwind from the polynya"; why is this variable (wind) not considered in your reconstruction?*
See main comment.

*Line 447: Insert "atmosphere" before "model".*
Modified as suggested.

*Line 450: "ice cores can be used" -> "it is tempting to use".*
Modified as suggested.

*Line 453: "downwind": this has not been shown.*
See above.

*Line 458: "simple average" of what?; "data assimilation": what data has been assimilated?*
This is the average of standardized surface mass records in the sector 50°W and 5°E and data assimilation constrained by the same records. This is now specified in the revised version.

*Line 462: "of the index": what index?*
This is the index of polynya activity. This is now specified in the revised version.

*Line 464: Add "Criscitiello et al., 2013"; see reference list in Ethan Campbell's comments.*
The reference has been added, as discussed in our response to Ethan Campbell's comments.

*Line 465: What does "these" refer to?*
It is the paleoclimate records mentioned in the beginning of the sentence. This is repeated in the revised version for clarity.

*Lines 470-471: Or much larger polynyas, or indeed ice embayments in the Weddell Sea, as often simulated (see e.g. Cheon et al., 2014; Kurtakoti et al., 2018).*
Much larger polynyas than the ones in the 1970's (or large ice embayments in winter) should have a larger signal and should thus be easier to detect with our methodology. However, we cannot determine if the fingerprint of those larger polynyas on snow accumulation over the continent seen in models is realistic as we have no clear equivalent from observations. The uncertainties would thus be very large for those kind of events. We propose to add in the manuscript a general point on 'polynya or ice embayments with a signal different from the polynya in the 1970's'.

Line 473: What does "few" mean? 2-3 times?
We have added in the revised version the mean number of polynya events for each reconstruction. For the reconstructions presented in the submitted manuscript, the mean number of years with open ocean

polynya for the whole period ranges from 1.8 to 4.7 years per century (criteria at 0.8 for the three reconstruction methods using the 5 long records).

315

Response Reviewer 2

We would like to thank the reviewer for the careful evaluation of our work and for the constructive comments. In the following, the reviewer's points are in italics, our response in plain text. When it is not specified otherwise, the line numbers refer to the revised version without track changes.

General Comments

*This article is an interesting and valuable contribution to our ability to determine the frequency of large, multi-year open-ocean polynyas in the Weddell Sea. The authors use a combination of continental observations and atmospheric model simulations to identify the potential signature from these open-ocean polynyas in continental ice cores located between 50°W and 0°E. The authors then use a series of high-resolution ice core records to estimate when large Weddell Sea open-ocean polynyas have occurred during the 800 yrs prior to the satellite era. There are substantial uncertainties associated with the reconstruction methods utilised here, which are clearly stated and discussed throughout the paper.*

*The paper presents novel techniques for identifying past open-ocean polynyas which are worth publishing. However, the uncertainties result in the majority of the conclusions being largely speculative. Therefore, this paper would benefit from a refocused discussion on the possible wider implications of the predicted polynya occurrence frequency, as these implications would help guide future work into corroborating or disproving the polynya frequency predicted by the authors.*

We agree with the reviewer that the uncertainties are large. This has been expanded in the revised version, in particular by presenting the reconstructions differently (see below) and by discussing more explicitly the perspective to reduce the uncertainties, as suggested, in new paragraphs of the final section (Discussion and Conclusions)

Specific Comments

*=> The introduction is very long, with substantial detail on polynya formation that seems superfluous to the focus and aims of the paper. If more discussion were to be added on the link between the formation mechanisms and the predicted occurrences in Figure 6 then this detail would become more relevant to the scope of the paper.*

As suggested by the reviewer, we have added in the final section (Discussion and Conclusions) new paragraphs (lines 477-504) to discuss the link between the mechanisms of formation and the predicted

occurrence of open ocean polynya. As suggested by the Editor, we have moved the corresponding parts of the introduction to this final paragraph. Specifically, this final section includes now a discussion of the wind changes, in particular how a connection between our reconstructions of polynya activity can be made with available reconstructions of the Southern Annular Mode (SAM; related to the intensity and the position of the Westerlies) for the past millennium. The link to oceanic processes is a bit more difficult to develop in the final section because of the small number of high resolution data available, but we have added an explanation of the potential impact of the reconstructed changes on the ocean state and how new information about past ocean changes could support, or refute some of the changes in polynya occurrence estimated from our reconstruction. Additionally, we have shortened the introduction to focus on the elements specifically relevant to the other sections of the manuscript.

*=> In both Figures 3a and 4 there are anomalies across large parts of the West Antarctic Peninsula and Amundsen-Bellingshausen Seas regions. The authors indicate in line 274 that these anomalies are not related to the Weddell Sea polynya. However, there is no further discussion of what is causing these anomalies. It is important to at least speculate as to what is causing these anomalies and, crucially, whether it is also responsible for the anomalies closer to the polynya that are currently being interpreted as a signal from the polynya itself.*

We agree that the sentence line 274 in the submitted version was a bit too short and that the topic deserves a more detailed explanation. We propose to change this line from 'it is unlikely that they are all related to the great Weddell Sea polynya formation' to ' it is unclear if they are all related to the great Weddell Sea polynya formation' and discuss further this point in section 4. Specifically, we have added, in the section devoted to the description of the fingerprint of the polynya, a justification as to why we consider that the higher accumulation in the section roughly between 50°W and 0° is a more robust signature of the open ocean polynya formation compared to other areas of the ice sheet (lines 335-346).

*=> Concerning the Hadley Centre data set used to drive the ECHAM5-wiso atmospheric model, it is not clear which years have been used in this study. Lines 189-190 suggest that data from 1871-2011 has been used whereas Figure 4 only seems to have the years 1958-2000. For either option there needs to be an appreciation of the limitations in using the Antarctic sea ice data from Rayner et al. (2003) to run the*

*model. For example, the Antarctic sea-ice extents for the years 1871-1927 are all duplications of the 1927 climatology.*

375 The Rayner et al. (2003 and updates) estimate of the Antarctic sea ice concentration has indeed clear limitations, with could potentially lead to large uncertainties in the ECHAM5-wiso results. The simulation covers the period 1871-2011, as mentioned in section 2.2. However, we only use the period 1958-2000 here to obtain a climatology of the model results that we compare with the years 1974-1976 corresponding to the great Weddell polynya existence. For clarity, this is now specified in the revised version when we

380 describe the simulation itself (lines 154-161). We do not use the earlier period, in particular the years 1871-1927 mentioned by the Reviewer when the uncertainties on the sea ice extent are particularly large. The uncertainties are already large before 1973 as no satellite-based imagery is directly included and the reconstruction is derived from various climatologies. We have chosen the years 1958-2000 as a reference to have a common period for all the datasets in Figures 2, 3 and 4. Nevertheless, our results for the region

385 of the ice sheet close to the Weddell Sea, and therefore more directly influenced by the polynya formation, are not very sensitive to the reference period chosen. This is illustrated in the figure below (Figure R2.1) where the reference period is 1979-2011 (when satellite information is available and the uncertainties on the Rainer et al. (2003) data set are lowest).

[Figure]

[Figure]

Figure R2.1: Anomaly of (a) annual mean temperature (°C), (b) precipitation (mm w.e./y), averaged over 1974-1976 compared to the period 1979-2011 in a simulation performed with ECHAM5-wiso. This figure is similar to Figure 4 of the main manuscript but with a different reference period.

*=> There should be some discussion of how changes in wind direction could influence the variability seen in Figures 6 and 7 as well as possible explanations for the hypothesised centennial-scale variability in line 432.*

As suggested, we have included a discussion of the possible role of wind changes, in particular related to the Southern Annular Mode, in the variability seen in our reconstruction. See also the response to the main comment above.

Technical Corrections

*Lines 55-57: statements on the ocean preconditioning should be referenced.*

References has been added.

*Line 229: replace "averages is" with "averages are".*

Corrected.

*Line 357: presumably should be 50°W not 50°E, as in line 368.*

Thank you for pointing out this mistake. It has been corrected in the revised version.

*Line 399: should be Figure 5 not Figure 4.*

This has been corrected.

*Lines 425-428: it is unclear whether this is referring to years when any single index is greater than 0.8/1 or when the average of all the indices is greater than 0.8/1.*

The discussion is written for any individual reconstruction. This is specified in the revised version.

*Figure 6: the use of "complete" and "all" to identify different time ranges is confusing for the reader.*

Instead of the reconstructions based on the 6 ice core records ('all') or only on the records that cover the entire studied period (i.e. 1250-now); 'complete'), we propose to estimate the uncertainties of the reconstructions by showing the reconstructions obtained when selecting five out of the 6 records in turn

for the assimilation. This is a more comprehensive and objective way to measure the uncertainties (e.g. Hakim et al., 2016). The word 'complete' and 'all" is thus not used anymore in the revised version.

*Figure 6: all the overlapping coloured records make it difficult to identify years with high values in all*
*the indices, especially in Figure 6a. The addition of arrows to indicate which years had an average index*
*value greater than 0.8 or 1 would be beneficial for analysing the changes in polynya frequency during*
*the last millennium.*

Figure 6 has been modified (as well as figure 7). First, we show the period 1850-1992 on figure 6 and the index for period 1250-1992 on the new figure 7. The full period is also presented in the new figures 8 and 9, the estimation on the number of events per 50-year intervals providing a strong smoothing allowing a complementary information compared to new figure 7. The interested reader can also check the exact value of the indices for each year in the table provided as supplementary material where the indices for the various reconstructions are given. For each figure (6,7,8,9), we first propose a panel with the reconstruction using the three methods (simple average and data assimilation using SPEAR_AM2 and SPEAR_LO models). We then propose one panel for each reconstruction in which the uncertainties are estimated by the range in the reconstruction from selecting only 5 out of the 6 ice core records (we exclude each record one-by-one), using the same methodology. This has the advantage of reducing the number of curves per panel and thus providing clearer figures. The first panel provides the uncertainty related to the method, while the other one represents the uncertainty related to the proxy selection for each method. We have also added horizontal lines on figures 6 and 7 for values of 0.8 and 1.0 to clearly see when each threshold is crossed (the other horizontal bars have been removed).

References

Hakim, G. J., J. Emile-Geay, E. J. Steig, D. Noone, D. M. Anderson, R. Tardif, N. Steiger, and W. A. Perkins (2016), 
[revised manuscript text omitted]